# Objaverse-XL: A Universe of 10M+ 3D Objects

**Matt Deitke**[†ψ]**, Ruoshi Liu**[γ]**, Matthew Wallingford**[ψ]**, Huong Ngo**[ψ]**, Oscar Michel**[†]**,**
**Aditya Kusupati**[ψ]**, Alan Fan**[ψ]**, Christian Laforte**[σ]**, Vikram Voleti**[σ]**, Samir Yitzhak Gadre**[γ]**,**
**Eli VanderBilt**[†]**, Aniruddha Kembhavi**[†ψ]**, Carl Vondrick**[γ]**, Georgia Gkioxari**[δ]**,**
**Kiana Ehsani**[†]**, \*Ludwig Schmidt**[†ψℓ]**, \*Ali Farhadi**[ψ]

[†]Allen Institute for AI   [ψ]University of Washington, Seattle   [γ]Columbia University
[σ]Stability AI   [δ]California Institute of Technology   [ℓ]LAION
*Equal Senior Contribution

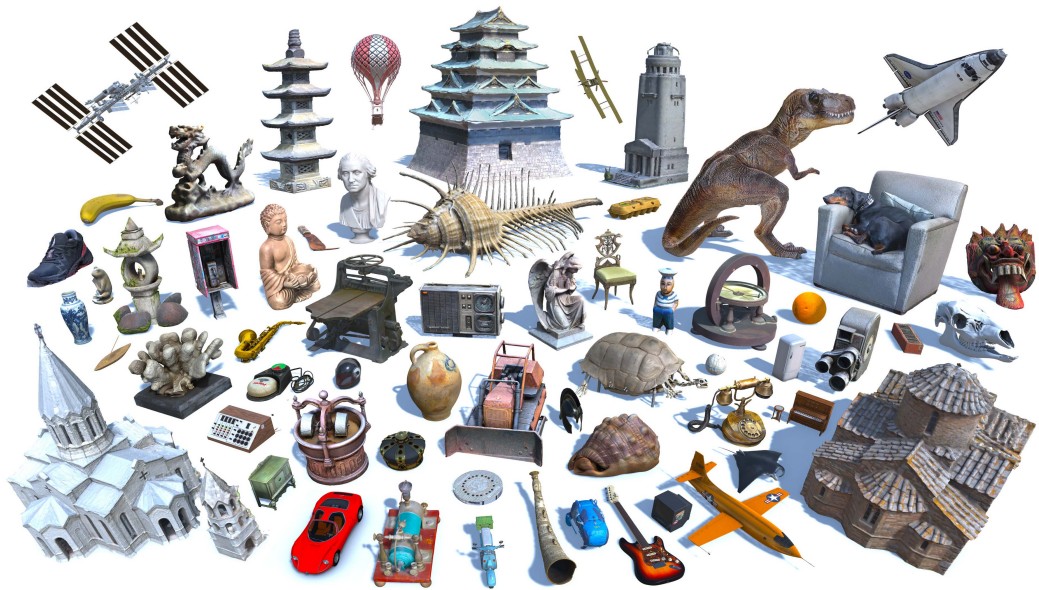

Figure 1: Objaverse-XL includes a diverse collection of 10M+ 3D objects from a variety of sources. Here, we show examples of objects in Objaverse-XL rendered in a scene.

## Abstract

Natural language processing and 2D vision models have attained remarkable proficiency on many tasks primarily by escalating the scale of training data. However, 3D vision tasks have not seen the same progress, in part due to the challenges of acquiring high-quality 3D data. In this work, we present Objaverse-XL, a dataset of over 10 million 3D objects. Our dataset comprises deduplicated 3D objects from a diverse set of sources, including manually designed objects, photogrammetry scans of landmarks and everyday items, and professional scans of historic and antique artifacts. Representing the largest scale and diversity in the realm of 3D datasets, Objaverse-XL enables significant new possibilities for 3D vision. Our experiments demonstrate the improvements enabled with the scale provided by Objaverse-XL. We show that by training Zero123 on novel view synthesis, utilizing over 100 million multi-view rendered images, we achieve strong zero-shot generalization abilities. We hope that releasing Objaverse-XL will enable further innovations in the field of 3D vision at scale.

37th Conference on Neural Information Processing Systems (NeurIPS 2023) Track on Datasets and Benchmarks.

# 1  Introduction

Scale has been paramount to recent advances in AI. Large models have produced breakthroughs in language comprehension and generation [1, 2], representation learning [3], multimodal task completion [4, 5], image generation [6, 7], and more. With an increasing number of learnable parameters, modern neural networks consume increasingly large volumes of data. As data has scaled up, the capabilities exhibited by models has dramatically increased.

Just a few years ago, GPT-2 [8] broke data barriers by consuming roughly 30 billion language tokens and demonstrated promising zero shot results on NLP benchmarks. Now, models such as Chinchilla [9] and LLaMA [10] consume trillions of web crawled tokens and easily surpass GPT-2 at benchmarks and capabilities. In computer vision, ImageNet [11], with 1 million images, was the gold standard for representation learning until scaling to billions of images, via web crawled datasets such as LAION-5B [12], produced powerful visual representations such as Contrastive Language-Image Pre-training (CLIP) [3]. The key to scaling up from millions of data points to billions and beyond has been the shift from assembling datasets manually to assembling them from diverse sources via the web.

As language and image data has scaled up, applications that require other forms of data have been left behind. Notable are applications in 3D computer vision, with tasks like 3D object generation and reconstruction, continue to consume small handcrafted datasets. 3D datasets such as ShapeNet [13] rely on professional 3D designers using expensive software to create assets, making the process tremendously difficult to crowdsource and scale. The resulting data scarcity has become a bottleneck for learning-driven methods in 3D computer vision. For instance, 3D object generation currently lags far behind 2D image generation, and current 3D generation approaches often still leverage models trained on large 2D datasets instead of being trained on 3D data from scratch. As demand and interest in AR and VR technologies goes up, scaling up 3D data is going to be increasingly crucial.

We introduce Objaverse-XL, a large-scale, web-crawled dataset of 3D assets. Advances in 3D authoring tools, demand, and photogrammetry, have substantially increased the amount of 3D data on the Internet. This data is spread across numerous locations including software hosting services like Github, specialized sites for 3D assets like Sketchfab, 3D printing asset sources like Thingiverse, 3D scanning platforms like Polycam, and specialized sites like the Smithsonian Institute. Objaverse-XL crawls such sources for 3D objects, providing a significantly richer variety and quality of 3D data than previously available, see Figure 1. Overall, Objaverse-XL comprises of over 10 million 3D objects, representing an order of magnitude more data than the recently proposed Objaverse 1.0 [14]. Objaverse-XL is two orders of magnitude larger than ShapeNet.

The scale and diversity of assets in Objaverse-XL significantly expands the performance of state-of-the-art 3D models. The recently proposed Zero123 [15] model for novel view synthesis, when pre-trained with Objaverse-XL, shows significantly better zero-shot generalization to challenging and complex modalities including photorealistic assets, cartoons, drawings and sketches. Similar improvements are also seen with PixelNerf which is trained to synthesize novel views given a small set of images. On each of these tasks, scaling pre-training data continues to show improvements from a thousand assets all the way up to 10 million, with few signs of slowing down, showing the promise and opportunities enabled with web scale data.

# 2  Related Work

**Pre-training Datasets.**   Massive datasets have a prevalent role in modern, data-driven AI as they have produced powerful and general representations when paired with large-scale training. In computer vision, ImageNet [11], introduced nearly 14 years ago, has become the standard pre-training dataset of state-of-the-art visual models in object detection [16, 17], instance segmentation [18, 19] and more. More recently, large image datasets, such as LAION-5B [12], have powered exciting advances in generative AI, such Stable Diffusion [7], and have given rise to new general-purpose vision and language representations with models like CLIP [3] and Flamingo [4]. More recently,

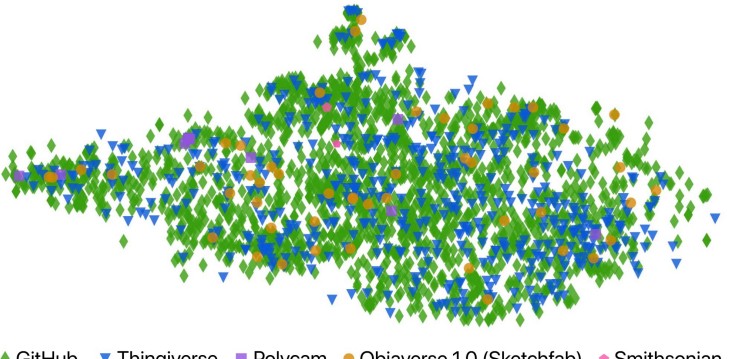

| Source | # Objects |
|---|---|
| IKEA [20] | 219 |
| GSO [21] | 1K |
| EGAD [22] | 2K |
| OmniObject3D [23] | 6K |
| PhotoShape [24] | 5K |
| ABO [25] | 8K |
| Thingi10K [26] | 10K |
| 3d-Future [27] | 10K |
| ShapeNet [13] | 51K |
| Objaverse 1.0 [14] | 800K |
| **Objaverse-XL** | **10.2M** |

◆ GitHub  ▼ Thingiverse  ■ Polycam  ● Objaverse 1.0 (Sketchfab)  ● Smithsonian

Figure 2: t-SNE projection of CLIP L/14 embeddings on a subset of rendered objects. Compared to Objaverse 1.0 (orange), Objaverse-XL more densely captures the distribution of 3D assets.

Table 1: Number of 3D models in common datasets. Objaverse-XL is over an order of magnitude larger than prior datasets.

Segment Anything (SAM) [28] introduced a dataset of one billion object masks used to train a model capable of segmenting any object from an image. In language understanding, datasets like Common Crawl [29] have culminated in unprecedented capabilities of large language models such as GPT-4 [2], which in turn power mainstream applications like ChatGPT. The impact of large datasets is undeniable. However, current efforts to collect massive datasets focus on image and language modalities. In this work we introduce and release publically a massive dataset of 3D objects, called Objaverse-XL. Given the promise of large datasets for 2D vision and language, we believe Objaverse-XL will accelerate research in large-scale training for 3D understanding.

**3D Datasets.** Existing 3D datasets have been instrumental in yielding significant findings in 3D over the years. ShapeNet [13] has served as the testbed for modeling, representing and predicting 3D shapes in the era of deep learning. ShapeNet provides a collection of 3D shapes, in the form of textured computer-aided design (CAD) models labeled with semantic categories from WordNet [30]. In theory, it contains 3M CAD models with textures. In practice, a small subset of 51K models is used after filtering by mesh and texture quality. Notwithstanding its impact, ShapeNet objects are of low resolution and textures are often overly simplistic. Other datasets, such as Amazon Berkeley Objects (ABO) [25], Google Scanned Objects (GSO) [21], ScanNet3D [31], Articulated Knowledge Base-48 (AKB-48) [32] and OmniObjects3D [23] for objects, and Matterport3D [33] and Habitat-Matterport 3D [34] for scenes, improve on the diversity or quality of their 3D models, by either scanning objects or using more advanced modeling techniques, but are significantly smaller in size with the largest constituting 15K 3D models. Recently, Objaverse 1.0 [14] introduced a 3D dataset of 800K 3D models with high quality and diverse textures, geometry and object types, making it 15× larger than prior 3D datasets. While impressive and a step toward a large-scale 3D dataset, Objaverse 1.0 remains several magnitudes smaller than dominant datasets in vision and language. As seen in Figure 2 and Table 1, Objaverse-XL extends Objaverse 1.0 to an even larger 3D dataset of 10.2M unique objects from a diverse set of sources, object shapes, and categories. We discuss Objaverse-XL and its properties in Section 3.

**3D Applications.** The potential of a massive 3D dataset like Objaverse-XL promises exciting novel applications in computer vision, graphics, augmented reality and generative AI. Reconstructing 3D objects from images is a longstanding problem in computer vision and graphics. Here, several methods explore novel representations [35, 36, 37, 38], network architectures [39, 40] and differentiable rendering techniques [41, 42, 43, 44, 45] to predict the 3D shapes and textures of objects from images with or without 3D supervision. All of the aforementioned projects experiment on the small scale ShapeNet. The significantly larger Objaverse-XL could pave the way to new levels of performance, and increase generalization to new domains in a zero-shot fashion. Over the past year, generative AI has made its foray into 3D. MCC [46] learns a generalizable representation with self-supervised learning for 3D reconstruction. DreamFusion [47] and later on Magic3D [48] demonstrated that 3D shapes could be generated from language prompts with the help of text-to-image models. Point-E [49] and Shape-E [50] also train for text-to-3D with the help of 3D models from an undisclosed source. Recently, Zero123 [15] introduced an image-conditioned diffusion model which generates novel

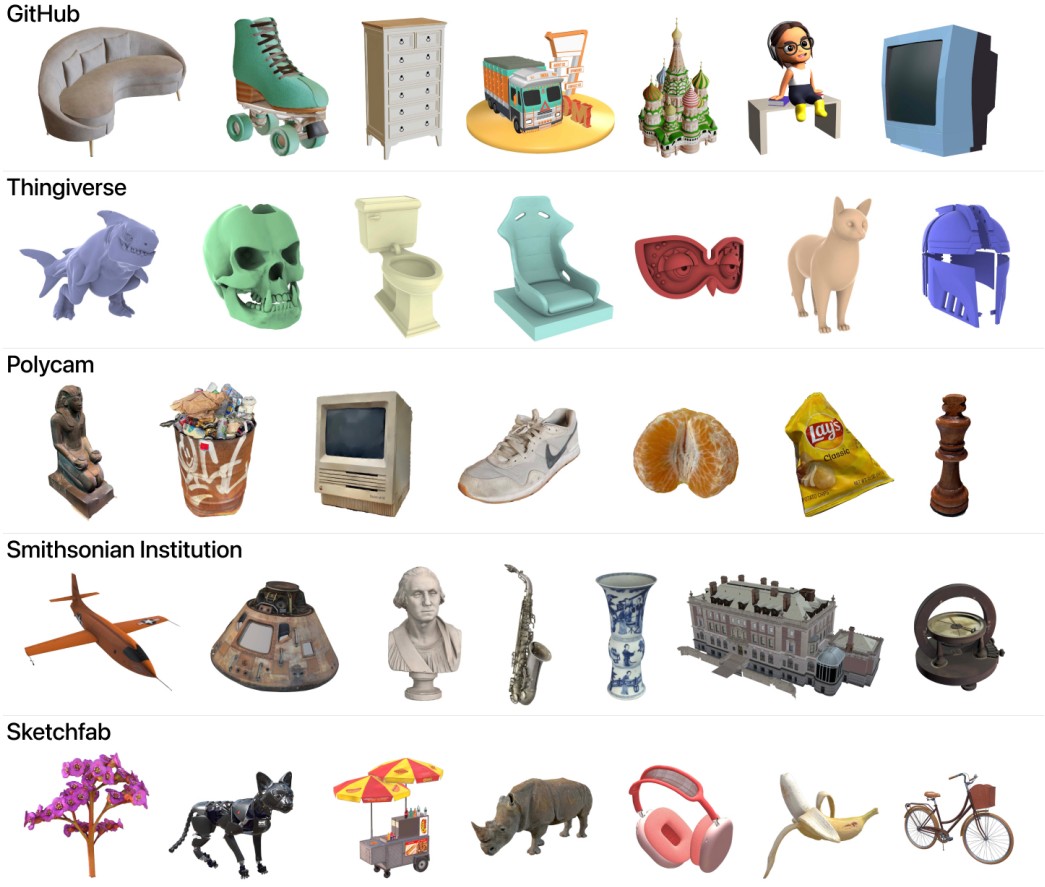

Figure 3: **Examples of 3D objects from various sources of Objaverse-XL** spanning GitHub, Thingiverse, Polycam, the Smithsonian Institution, and Sketchfab. Objects from Thingiverse do not include color information, so each object's primary color is randomized during rendering.

object views and is trained on Objaverse 1.0. Stable Dreamfusion [51] replaces the text-to-image model in DreamFusion with the 3D-informed Zero123 and shows improved 3D generations. Recent findings in AI and scaling laws [52, 9] suggest that both generative and predictive models benefit from larger models and larger pre-training datasets. For 3D, Objaverse-XL is by far the largest 3D dataset to date and has the potential to facilitate large-scale training for new applications in 3D.

## 3  Objaverse-XL

Objaverse-XL is a web scale 3D object dataset composed of a highly diverse set of 3D data sources on the internet. In this section, we discuss the sources, metadata of the objects, and provide an analysis of the objects.

### 3.1  Composition

Objaverse-XL is composed of 3D objects coming from several sources, including GitHub, Thingiverse, Sketchfab, Polycam, and the Smithsonian Institution. We detail each source below.

**GitHub** is a popular online platform for hosting code. We index 37M public files that contain common 3D object extensions; in particular, `.obj`, `.glb`, `.gltf`, `.usdz`, `.usd`, `.usda`, `.fbx`, `.stl`, `.dae`, `.ply`, `.abc`, and `.blend`. These extensions were chosen as they are best supported in Blender, which we use to render 2D images of the 3D objects. We only index objects that come from "base"

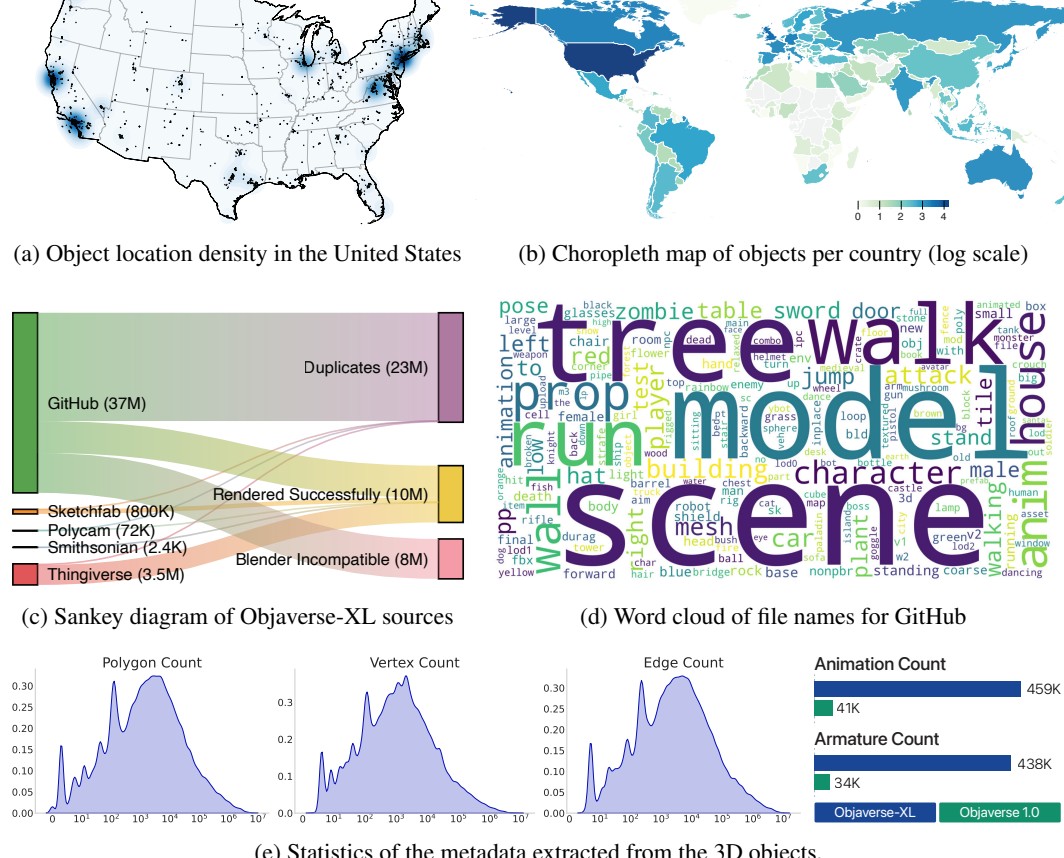

(a) Object location density in the United States

(b) Choropleth map of objects per country (log scale)

(c) Sankey diagram of Objaverse-XL sources

(d) Word cloud of file names for GitHub

(e) Statistics of the metadata extracted from the 3D objects.

Figure 4: **Analysis of metadata from Objaverse-XL.** Locations of geotagged objects in (a) the United States and (b) around the world. (c) Various sources and their contribution to Objaverse-XL. (d) Frequency of filenames of GitHub objects. (e) Further statistics of collected 3D objects.

GitHub repositories (*i.e.* non-forked repos, excluding forks that had more stars than the original repo). In total, the files come from over 500K repositories.

Across all of Objaverse-XL, objects are deduplicated by file content hash, which removes approximately 23 million files. Among the remaining files, we were able to import and successfully render 5.5 million of those files. Files that were not successfully rendered were either caused by import compatibility issues (*i.e.* FBX ASCII files are not natively importable to Blender), no meshes are in the files, or the file is not a valid 3D file (*e.g.* an `.obj` file may be a C compiler file instead of a Wavefront Object file). Moving forward, we expect a solution for converting 3D file formats into a consolidated representation may yield several million more unique 3D objects.

**Thingiverse** is a platform for sharing objects most commonly used for 3D printing. We index and download around 3.5 million objects from the platform, which are predominantly released under Creative Commons licenses. The vast majority of the objects are STL files, which are often watertight meshes that are untextured, and serve as useful data for learning a shape prior. During rendering, we randomize the colors to broaden the distribution of the images.

**Sketchfab** is an online platform where users can publish and share 3D models, encompassing a broad variety of categories. The data sourced from Sketchfab for our project is specifically from Objaverse 1.0, a dataset of 800K objects consisting of Creative Commons-licensed 3D models. Each model is distributed as a standardized GLB file. The 3D models are freely usable and modifiable, covering an array of object types, from real-world 3D scans to intricate designs created in 3D software.

**Polycam** is a 3D scanning mobile application designed to facilitate the acquisition and sharing of 3D data. One of its salient features is the *explore* functionality, which enables members of the user

community to contribute their 3D scans to a publicly accessible database. In the context of our dataset, we focus specifically on the subset of objects within the explore page that are designated as savable. These savable objects are governed by a Creative Commons Attribution 4.0 International License (CC-BY 4.0). We indexed 72K objects that were marked as savable and licensed under a CC-BY 4.0 license. Following deduplication, we obtain 71K unique objects.

**Smithsonian 3D Digitization** is a project by the Smithsonian Institution dedicated to digitizing their vast collection of historical and cultural artifacts. The project has provided us with a set of 2.4K models, all licensed under a CC0 license, which signifies that these works are fully in the public domain and free for use without any restrictions. The objects in this collection are primarily scans of real-world artifacts. Each model is distributed in a standardized compressed GLB format.

## 3.2 Metadata

Each object comes with metadata from its source, and we also extract metadata from it in Blender and from its CLIP ViT-L/14 features. We describe the metadata acquisition process below.

**Source Metadata.** From the source, we often get metadata such as its popularity, license, and some textual description. For example, on GitHub, the popularity is represented by the stars of the object's repository and the file name serves as the object's textual pair.

**Blender Metadata.** For each object that we render, we obtain the following metadata for it: `sha256`, `file-size`, `polygon-count`, `vertex-count`, `edge-count`, `material-count`, `texture-count`, `object-count`, `animation-count`, `linked-files`, `scene-dimensions`, and `missing-textures`. During rendering, for objects that have a missing texture, we randomize the color of that texture. Figure 4 shows some charts extracted from the metadata, including density plots over the number of polygons, vertex counts, and edge counts.

**Animated Objects.** From the Blender metadata, we find that the number of animated objects and those with armature (a digital skeleton used to animate 3D models) significantly increases from Objaverse 1.0 to Objaverse-XL. Figure 4e (right) shows a bar chart of the increase, specifically from 41K to 459K animated objects and from 34K to 438K objects with armature.

**Model Metadata.** For each object, we extract its CLIP ViT-L/14 [3] image embedding by averaging the CLIP embedding from 12 different renders of the object at random camera positions inside of a hollow sphere. We use the CLIP embeddings to predict different metadata properties, including aesthetic scores, not safe for work (NSFW) predictions, face detection, and for detecting holes in the photogrammetry renderings. Section D.2 provides more details on the analysis.

## 3.3 Analysis

**NSFW annotations.** Most data sources used for the creation of Objaverse-XL already have either a strict NSFW policy or strong self-filtering. However, owing to the web scale of Objaverse-XL we performed NSFW filtering using the rendered images of the objects. Each 3D object is rendered in 12 random views and each rendered image is passed through an NSFW classifier trained on the NSFW dataset introduced in LAION-5B [12] by **(author?)** [53] using the CLIP ViT-L/14 [3] features. After careful analysis and manual inspection, we marked a rendered image as NSFW if it has an NSFW score above $0.9$ and a 3D object is marked as NSFW if at least 3 rendered images are deemed to be NSFW. Overall, only $815$ objects out of the 10M are filtered out as NSFW objects. Note that the high threshold and multi-view consistency are needed due to the distribution shift between LAION-5B and Objaverse-XL along with NSFW classification of certain viewpoint renders of harmless 3D objects.

**Face detection.** We analyze the presence of faces in Objaverse-XL using a detector trained by **(author?)** [53]. Like NSFW filtering, we count the objects where at least 3 images contain a detected face. Out of 10M assets, we estimate 266K objects include faces. However, unlike most web-scale datasets, the faces present in Objaverse-XL often come from the scans of dolls, historical sculptures, and anthropomorphic animations. Hence, there are less privacy concerns with most of these objects.

**Photogrammetry hole detection.** When scanning 3D objects, if the back or bottom of the object is not scanned, rendering from various viewpoints may contain holes, leading to a "bad" render image.

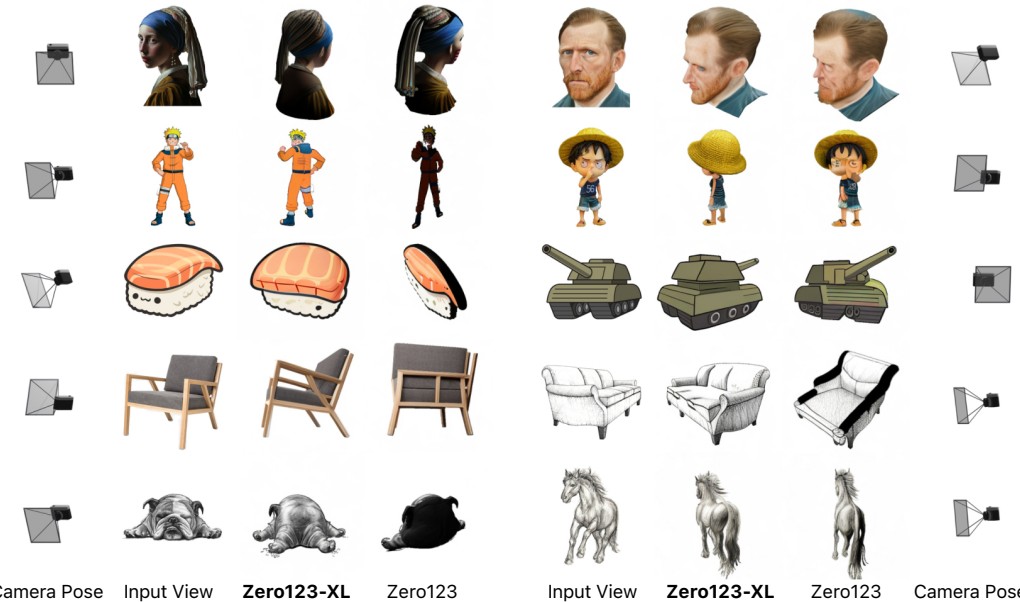

| Camera Pose | Input View | **Zero123-XL** | Zero123 | | Input View | **Zero123-XL** | Zero123 | Camera Pose |

Figure 5: **Novel view synthesis on in-the-wild images.** Comparison between Zero123-XL trained on Objaverse-XL and Zero123 trained on Objaverse. Starting from the input view, the task is to generate an image of the object under a specific camera pose transformation. The camera poses are shown beside each example. Significant improvement can be found by training with more data, especially for categories including people (**1st row**), anime (**2nd row**), cartoon (**3rd row**), furniture (**4th row**), and sketches (**5th row**). Additionally, viewpoint control is significantly improved (see **2nd row**).

For example, a non-trivial number of Polycam 3D objects lack the information from the "back side". In most cases, images that are rendered from back-side viewpoints are noisy, low-fidelity, or contain holes. To analyze "bad rendering" at scale, we manually annotated 1.2K Polycam renders as "good" (label 1) or "bad" (label 0). We trained a "bad render" classifier (2-layer MLP) on top of the CLIP ViT-L/14 features of the rendered images; this classifier achieves a cross-validation accuracy of over $90\%$ with a "render score" threshold of $0.5$. Overall, out of 71K Polycam objects with 12 renders each, we found that $38.20\%$ renders are "bad", with 58K objects having at least 2 bad renders.

## 4 Experiments

### 4.1 Novel View Synthesis with Zero123-XL

Generating 3D assets conditioned on in-the-wild 2D images has remained a challenging problem in computer vision. A crucial lesson learned from large language models is that pretraining on simple and easily scalable tasks, such as next word prediction, leads to consistently improved performance and the emergence of zero-shot abilities. An analogous approach in 3D vision is to predict a novel view of an object from an input view. Zero123 [15] recently proposed a view-conditioned diffusion model to perform this task, where the weights of the diffusion model are initialized from Stable Diffusion to leverage its powerful zero-shot image generation abilities. Zero123 used objects in

| Zero123-XL | PSNR ($\uparrow$) | SSIM ($\uparrow$) | LPIPS ($\downarrow$) | FID ($\downarrow$) |
|---|---|---|---|---|
| Base | 18.225 | 0.877 | 0.088 | 0.070 |
| w/ Alignment Finetuning | **19.876** | **0.888** | **0.075** | **0.056** |

Table 2: **Effect of high-quality data finetuning on Zero123-XL.** When evaluated zero-shot on Google Scanned Objects [21], a model finetuned on a high-quality alignment subset of Objaverse-XL significantly outperforms the base model trained only on Objaverse-XL.

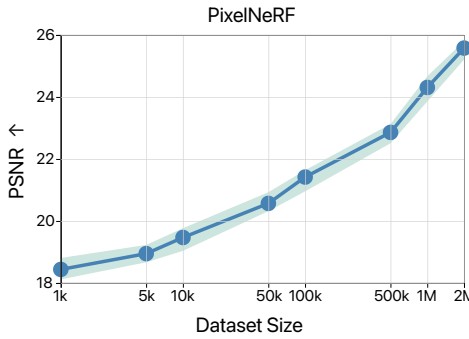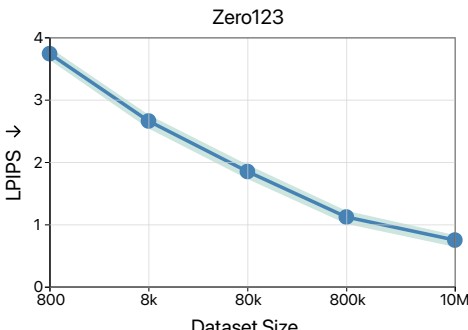

Figure 6: **Novel view synthesis at scale. Left**: PixelNeRF [40] trained on varying scales of data and evaluated on a held-out subset of Objavserse-XL. **Right**: Zero123 [15] trained on varying scales of data and evaluated on a zero-shot dataset. Note that the 800K datapoint is Zero123 and the 10M datapoint is Zero123-XL. The synthesis quality consistently improves with scale. LPIPS is scaled up 10 times for visualization.

Objaverse 1.0 to render input and novel view pairs as the training dataset. We use this framework to create *Zero123-XL*, which is the same approach except trained on the much larger Objaverse-XL instead. As shown in [15], the pretrained view-conditioned diffusion model can also be plugged into a score distillation framework such as DreamFusion [47] or SJC [54] to obtain a 3D assets.

**Zero-shot Generalization.**   We found that training Zero123 on Objaverse-XL achieves significantly better zero-shot generalization performance than using Objaverse 1.0. Figure 5 shows examples from categories of data commonly known to be challenging for baseline systems, including people, cartoons, paintings, and sketches. For example, in both of the examples shown in 2nd and 3rd rows of the first column, Zero123 interprets the input image as a 2D plane and performs a simple transformation similar to a homography transformation. In comparison, Zero123-XL is able to generate novel views that are more consistent with the input view. Additionally, Zero123-XL is able to generate novel views from sketches of objects while keeping the original style as well as object geometric details. These examples show the effectiveness of dataset scaling for zero-shot generalization in 3D.

**Improvement with Scale.**   We further quantitatively evaluate the novel view synthesis performance on Google Scanned Objects dataset [21]. As shown in Figure 6, the rvisual similarity score [55] between the predicted novel view and the ground truth view continues to improve as the dataset size increases.

**Alignment Finetuning.**   InstructGPT [56] shows that large-scale pretraining does not directly lead to a model aligned with human preferences. More recently, LIMA [57] shows that finetuning a pretrained model on a curated subset with high-quality data can achieve impressive alignment results. We adopted a similar approach here by selecting a high-quality subset of Objaverse-XL that contains 1.3 million objects. Selection is done by defining proxy estimation of human preference based on heuristics including vertex count, face count, popularity on the source website, and source of data, among other metrics. After pretraining the base model on the entire Objaverse-XL, we finetune Zero123-XL on the alignment subset with a reduced learning rate and performed an ablation study to evaluate the effect of alignment finetuning. Table 2 shows that alignment finetuning leads to significant improvement in zero-shot generalization performance. Please refer to Appendix A for more implementation details regarding our model and experiments.

## 4.2   Novel View Synthesis with PixelNeRF

Synthesizing objects and scenes from novel views is a long-standing challenge. Notably, neural radiance fields [38] have shown impressive capabilities in rendering specific scenes from novel views. However, these methods require dozens of views of an individual scene, and can only synthesize views from the particular scene they were trained for. More recent methods [58, 59, 60, 40] have been proposed for constructing NeRF models that generalize across scenes with few input images. Due to the challenging nature of obtaining the necessary camera parameters for training, such methods have

traditionally been trained on small scale data sets. With the Objaverse-XL data, we train a PixelNeRF model on over two million objects, magnitudes of more data than has previously been used. We find that PixelNeRF generalizes to novel scenes and objects significantly better and performance improves consistently with scale (Figure 6 and Table 3).

**Improvement with Scale.** We train PixelNeRF models conditioned on a single input image at varying scales of data (Figure 6) and evaluate on a held out set of Objaverse-XL objects. We find that novel view synthesis quality consistently improves with more objects even at the scale of 2 million objects and 24 million rendered images.

**Generalization to Downstream Datasets.** Similar to pretraining in 2D vision and language, we observe that pretraining on Objaverse-XL with PixelNeRF improves performance when fine-tuning to other datasets such as DTU [61] and ShapeNet [13] (Table 3). We pretrain and fine-tune the model conditioned on a single input view and report the peak signal-to-noise ratio (PSNR).

| PixelNeRF | DTU [61] | ShapeNet [13] |
|---|---|---|
| Baseline [40] | 15.32 | 22.71 |
| w/ Objaverse-XL | **17.53** $\pm.37$ | **24.22** $\pm.55$ |

Table 3: **Comparison (PSNR (↑)) of PixelNeRF trained from scratch vs. fine-tuned from Objaverse-XL.** Performance significantly improves from pretraining on the large-scale corpus.

## 5   Limitations and Conclusion

**Limitations.** While Objaverse-XL is more than an order of magnitude larger than its predecessor, Objaverse 1.0, it is still orders of magnitude smaller than modern billion-scale image-text datasets. Future work may consider how to continue scaling 3D datasets and make 3D content easier to capture and create. Additionally, it may not be the case that all samples in Objaverse-XL are necessary to train high performance models. Future work may also consider how to choose datapoints to train on. Finally, while we focus on generative tasks, future work may consider how Objaverse-XL can benefit discriminative tasks such as 3D segmentation and detection.

**Conclusion.** We introduce Objaverse-XL, which is comprised of 10.2M 3D assets. In addition to documenting Objaverse-XL's unprecedented scale and sample diversity, we demonstrate the potential of Objaverse-XL for downstream applications. On the task of zero-shot novel view synthesis, we establish empirically promising trends of scaling dataset size, while keeping the model architecture constant. We hope Objaverse-XL will provide a foundation for future work in 3D.

## Acknowledgements

We would like to thank Stability AI for compute used to train the experiments and LAION for their support. We would also like to thank Luca Weihs, Mitchell Wortsman, Romain Beaumont, Vaishaal Shankar, Rose Hendrix, Adam Letts, Sami Kama, Andreas Blattmann, Kunal Pratap Singh, and Kuo-Hao Zeng for their helpful guidance and conversations with the project. Finally, we would like to thank the teams behind several open-source packages used throughout this project, including Blender [62], PyTorch [63], PyTorch Lightning [64], D3 [65], Matplotlib [66], NumPy [67], Pandas [68], Wandb [69], and Seaborn [70]. Following the NeurIPS guidelines, we would also like to acknowledge the use of LLMs for helping revise some text and general coding assistance. Finally, we would also like to thank and acknowledge the content creators who contributed to the dataset.

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
