# A    Implementation Details

## A.1    Zero123-XL

A batch size of 2048 is used during training with a learning rate of 1e-4. Different from the original paper [15], we performed a second-stage finetuning with a smaller learning rate of 5e-5 on a high-quality subset of Objaverse-XL selected with dataset metadata. The first stage was trained for 375K iterations and the second stage is trained for 65K iterations. For dataset scaling experiment whose results are shown in 6, datasets with size below 800K are randomly sampled subsets from Objaverse 1.0. We keep the rest of the setting consistent with the original paper [15]. Training the Zero123-XL model was done on 256 NVIDIA A100s over the course of 4 days, for a total of around 25K GPU hours. Rendering the objects was done for around 1 week on 48 NVIDIA T4 GPUs, totaling around 8K GPU hours. Both training and rendering were conducted using AWS.

## A.2    PixelNeRF

We use the official implementation of PixelNeRF provided at `https://github.com/sxyu/pixel-nerf`. We trained the PixelNeRF models on 8 NVIDIA A100's over the course of 2 days with a total batch size of 256 for 200 epochs. We used a constant learning rate of 1e-4. We added gradient clipping to the original implementation as we found the loss would destabilize throughout training otherwise. We trained the model with 1 input view for the experiments reported in the main paper. For the train and validation sets we randomly split the data from the main Objaverse pool without filtering for aesthetics.

To train PixelNeRF on Objaverse-XL we render the meshes in Blender. For each object we render 12 images of the object, randomly sampled from the unit sphere with the object centered in the camera view. Each model is normalize to a bounding cube. For evaluation, we randomly sample 100,000 objects and 1.2 million renders disjoint from the training set.

# B    Societal Impacts

We believe that models such as Zero123-XL, and those trained on Objaverse-XL, will enhance the ease of 3D content creation, enabling broader accessibility for individuals and businesses to participate. With the rise of mixed-reality glasses such as Apple's Vision Pro, the demand for 3D content is surging. Creating 3D content with tools such as Blender and Maya is time consuming and difficult, with tasks such as modeling a mere sofa requiring a day's effort from an expert 3D artist. We hope that by making 3D development faster and more accessible, it will enable many future applications in robotics simulation, AR/VR, games, education, architecture, manufacturing, healthcare, medicine, and biology, among many other industries.

Openly accessible datasets, such as Objaverse-XL, serve as a crucial component for the democratization of knowledge to the AI community. Such resources allow others to access and utilize tools that might have been previously reserved for a select few organizations. Additionally, openly available datasets make it much easier for the AI community to verify the merits of different models and training algorithms, as it enables different organizations to use the same training data. Moreover, The inherent transparency of openly available datasets cultivate a foundation of trust. As these datasets are open to scrutiny, they invite collective oversight, ensuring that the methods and sources of data collection are held to high standards.

However, we also acknowledge that there are several potential negative societal impacts with the release of Objaverse-XL. One concern is the development hyper-realistic fake content, such as deepfakes, which can be harnessed for misinformation campaigns or identity theft. The automation capabilities of AI may diminish the demand for human roles in areas like 3D modeling, subtly altering the employment landscape. An over-reliance on AI for generating 3D objects may also lead to an oversaturation of similar designs, potentially suppressing human creativity. Lastly, despite the democratizing intent of openly accessible datasets, the primary beneficiaries might often be those with more substantial resources or technological infrastructure that can launch large-scale training runs, inadvertently perpetuating economic disparities.

## C  Additional Zero123-XL Comparisons

Figures 7-18 show additional comparisons between Zero123-XL and Zero123. Overall, Zero123-XL shows better generalization than Zero123 by both better following the camera transformation and generating more plausible outputs. Sources for the images are available here.

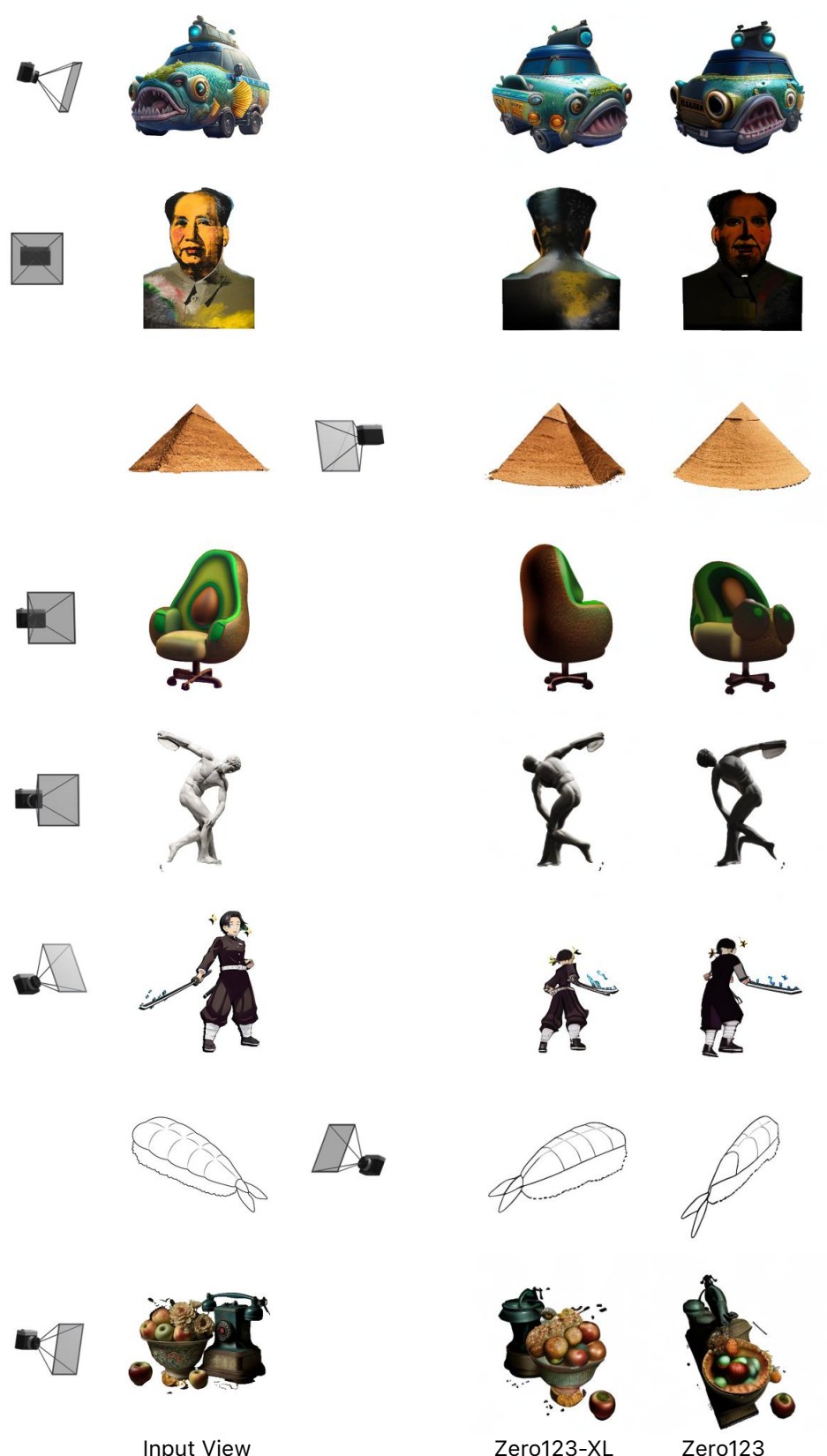

Input View            Zero123-XL      Zero123

Figure 7: Additional examples comparing the outputs of Zero123-XL and Zero123 under different camera transformations.

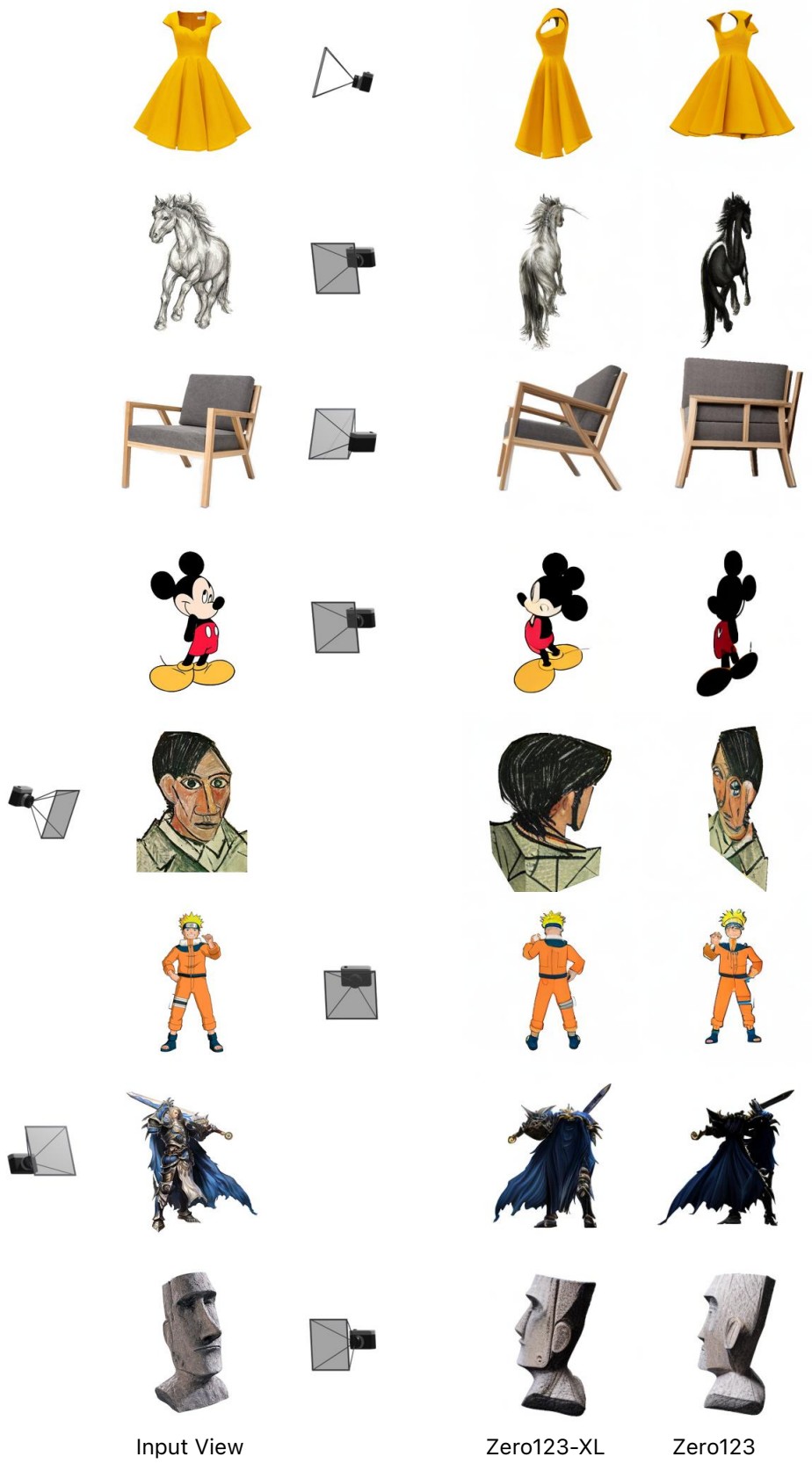

Input View        Zero123-XL     Zero123

Figure 8: Continuation of additional examples comparing Zero123-XL and Zero123.

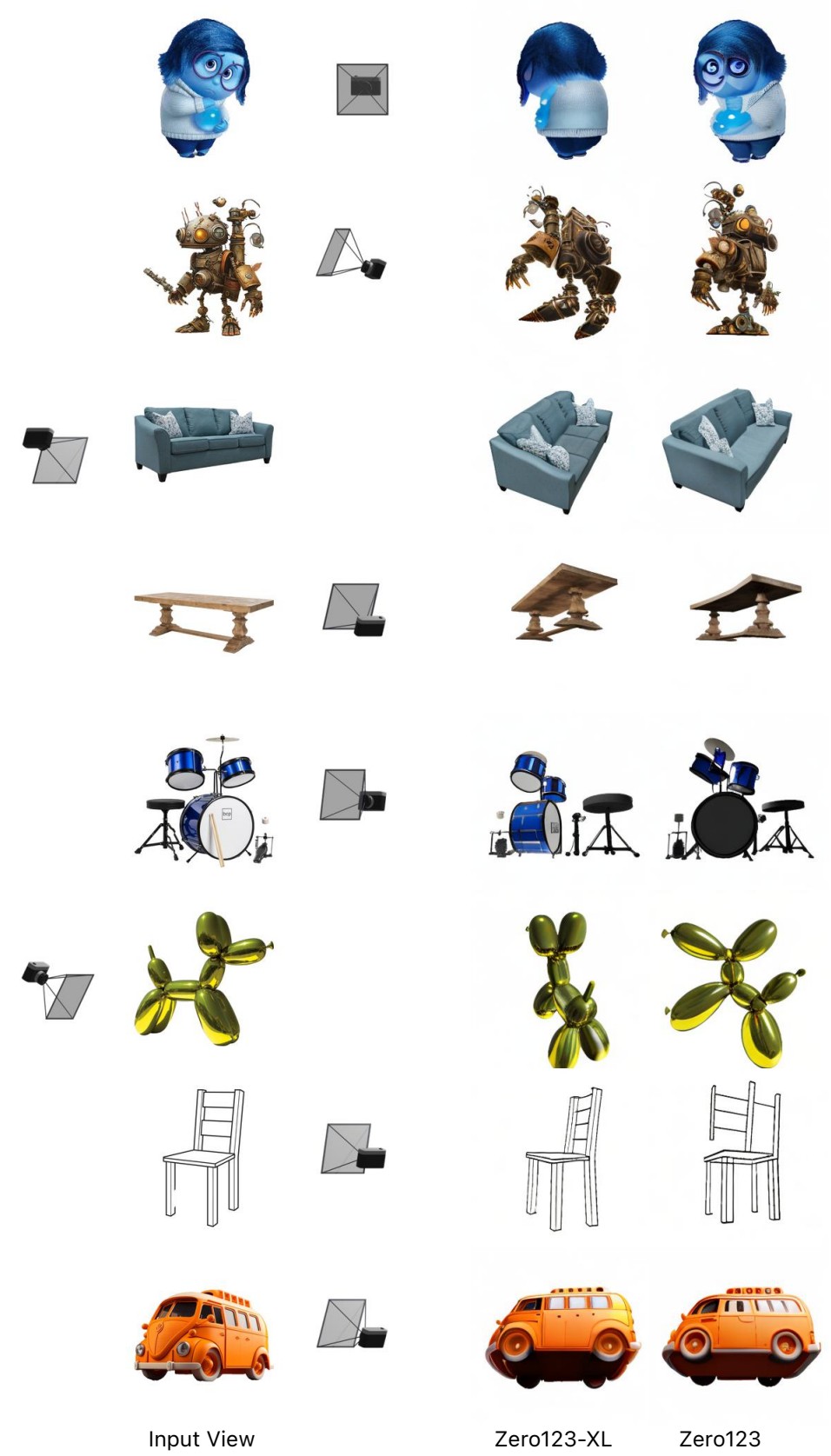

Input View         Zero123-XL    Zero123

Figure 9: Continuation of additional examples comparing Zero123-XL and Zero123.

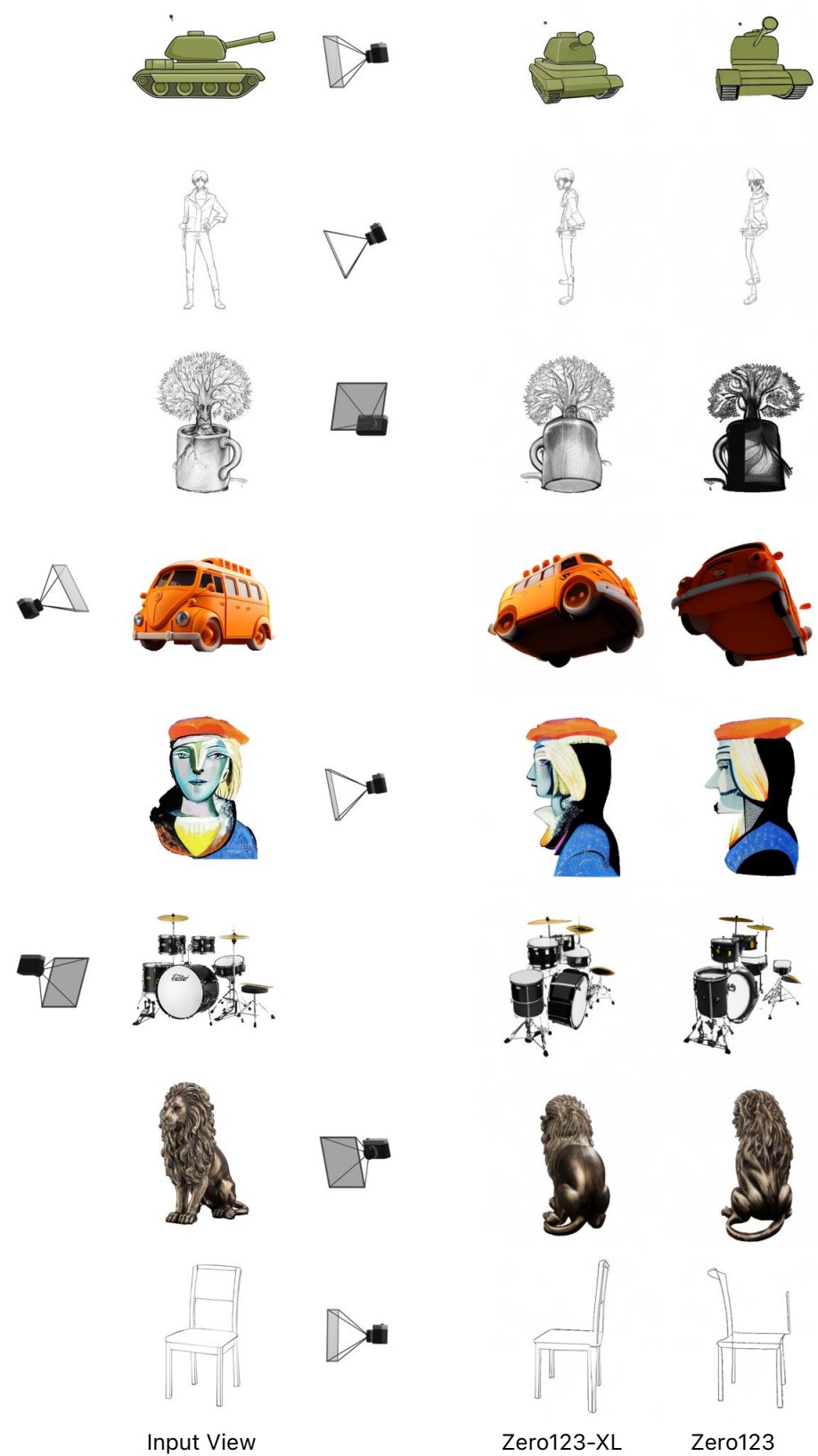

Input View       Zero123-XL   Zero123

Figure 10: Continuation of additional examples comparing Zero123-XL and Zero123.

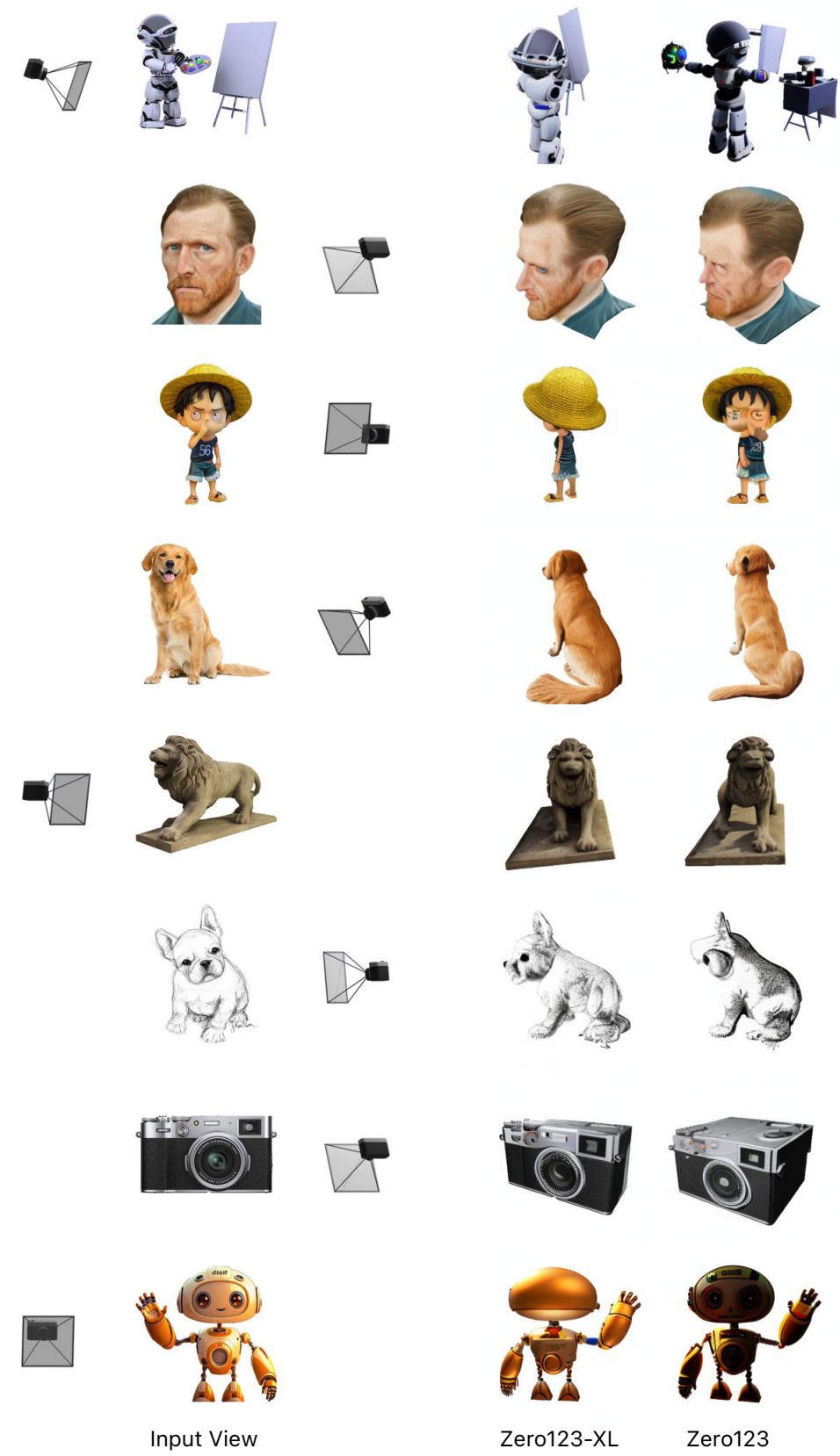

Input View       Zero123-XL  Zero123

Figure 11: Continuation of additional examples comparing Zero123-XL and Zero123.

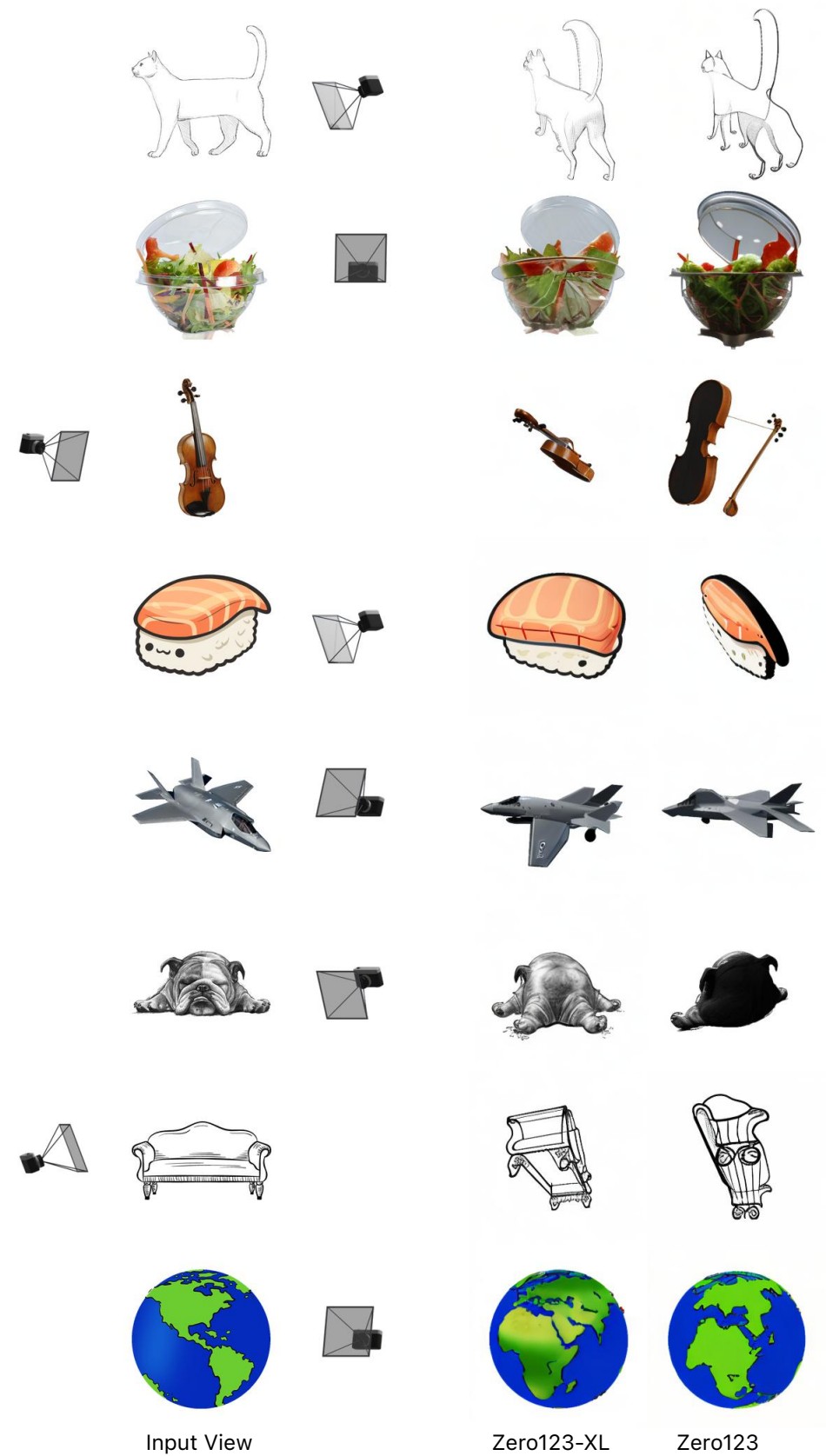

Input View      Zero123-XL  Zero123

Figure 12: Continuation of additional examples comparing Zero123-XL and Zero123.

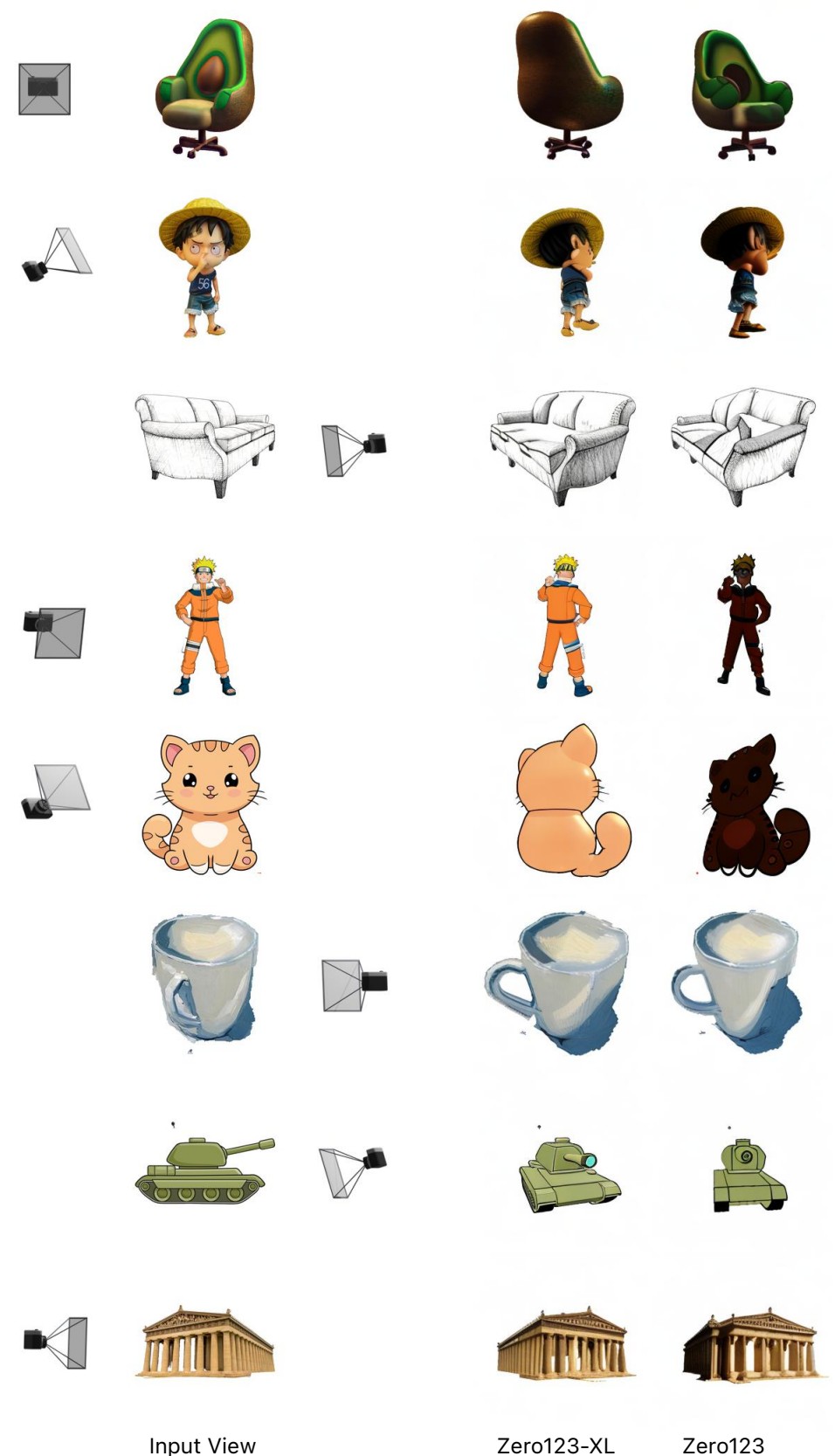

Input View                Zero123-XL        Zero123

Figure 13: Continuation of additional examples comparing Zero123-XL and Zero123.

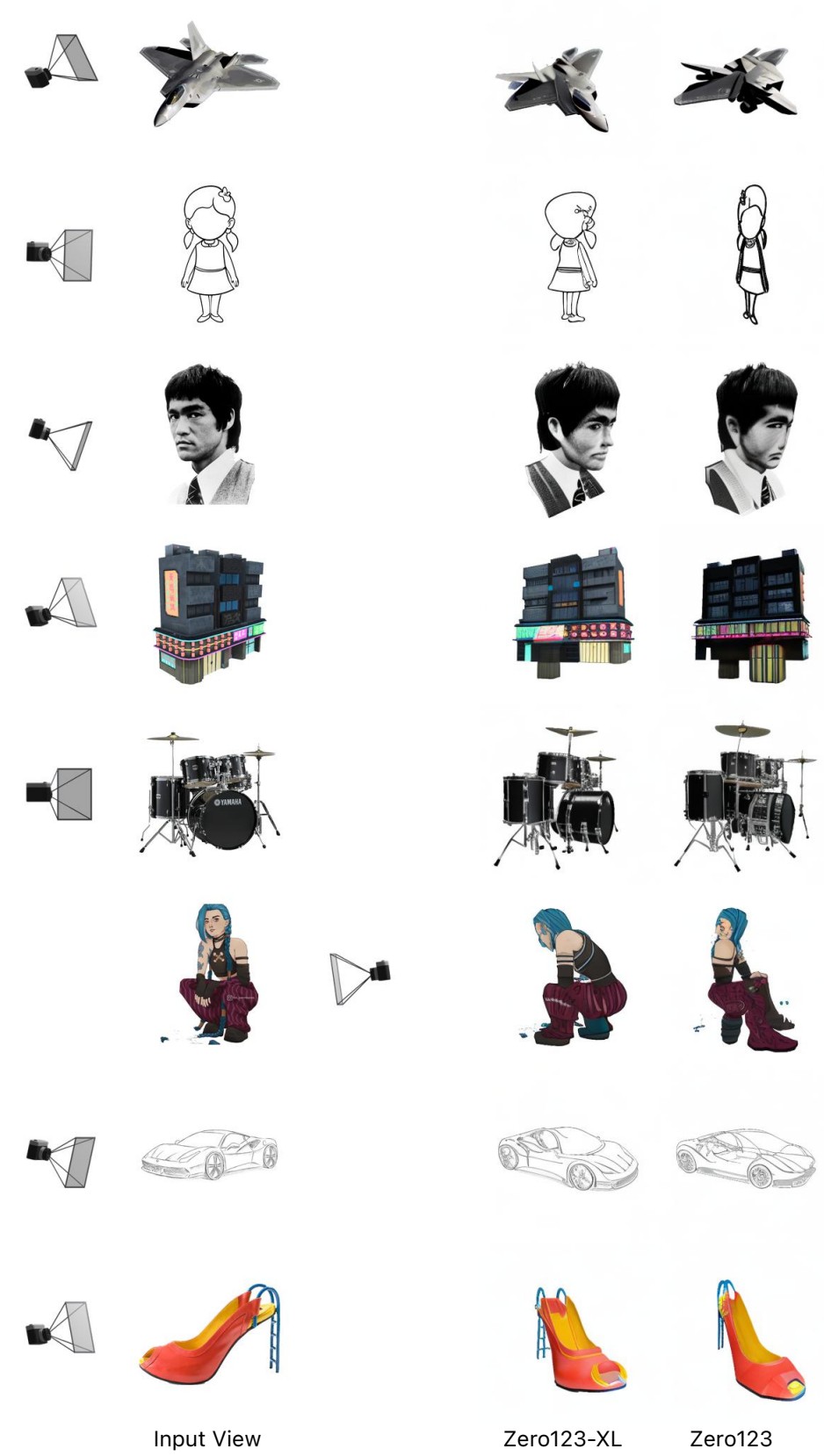

Input View       Zero123-XL   Zero123

Figure 14: Continuation of additional examples comparing Zero123-XL and Zero123.

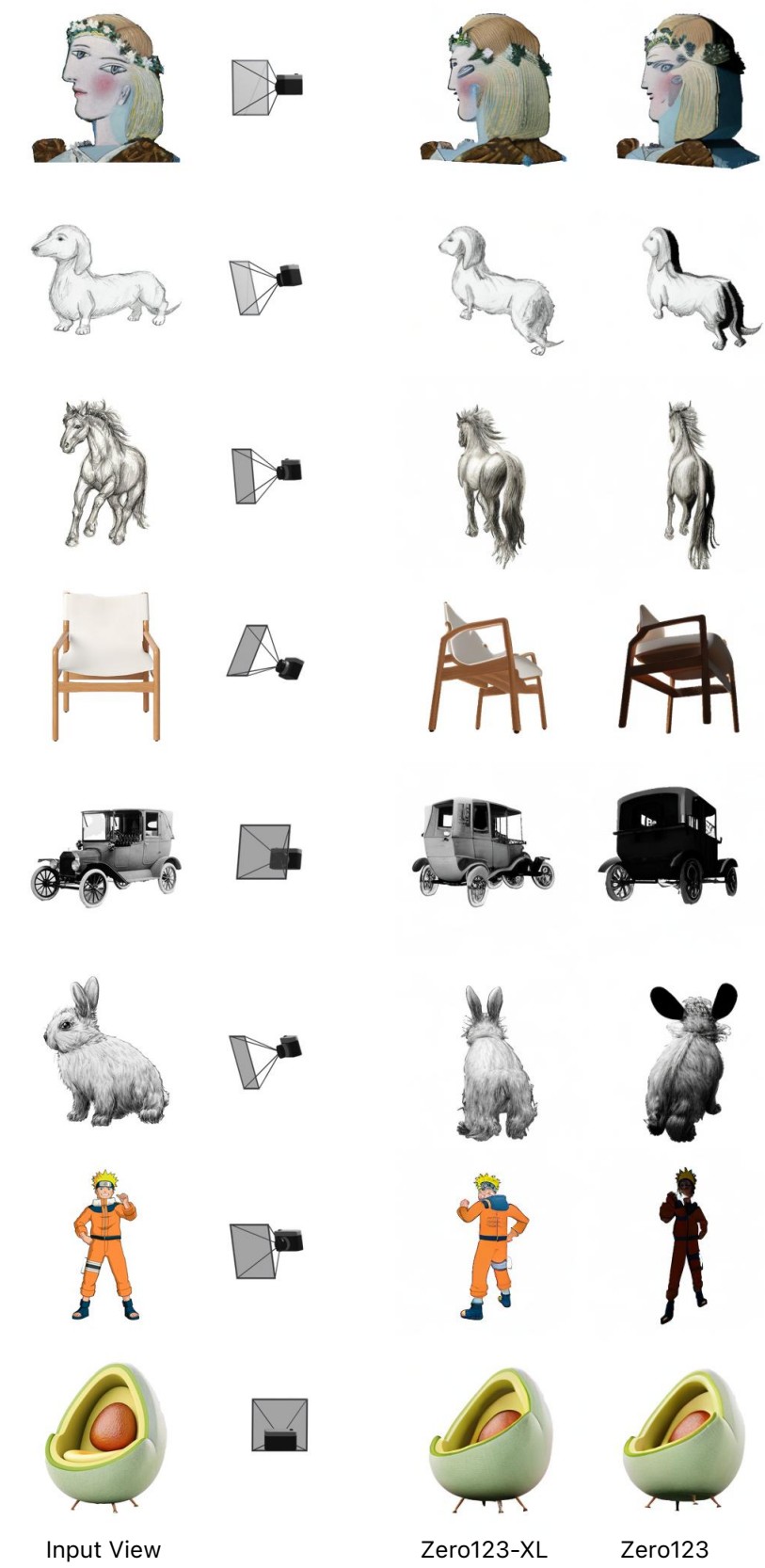

Input View        Zero123-XL   Zero123

Figure 15: Continuation of additional examples comparing Zero123-XL and Zero123.

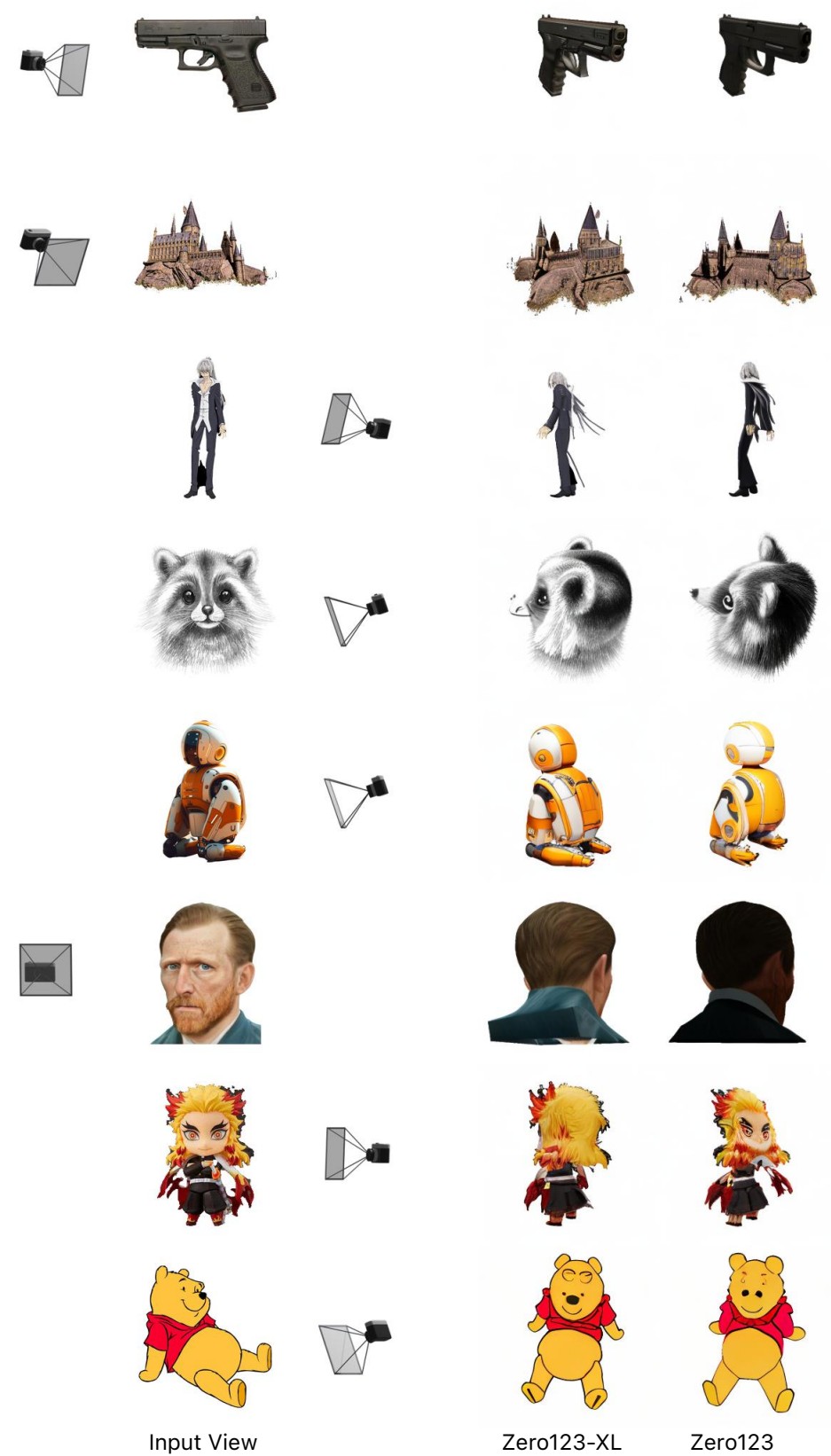

Input View        Zero123-XL     Zero123

Figure 16: Continuation of additional examples comparing Zero123-XL and Zero123.

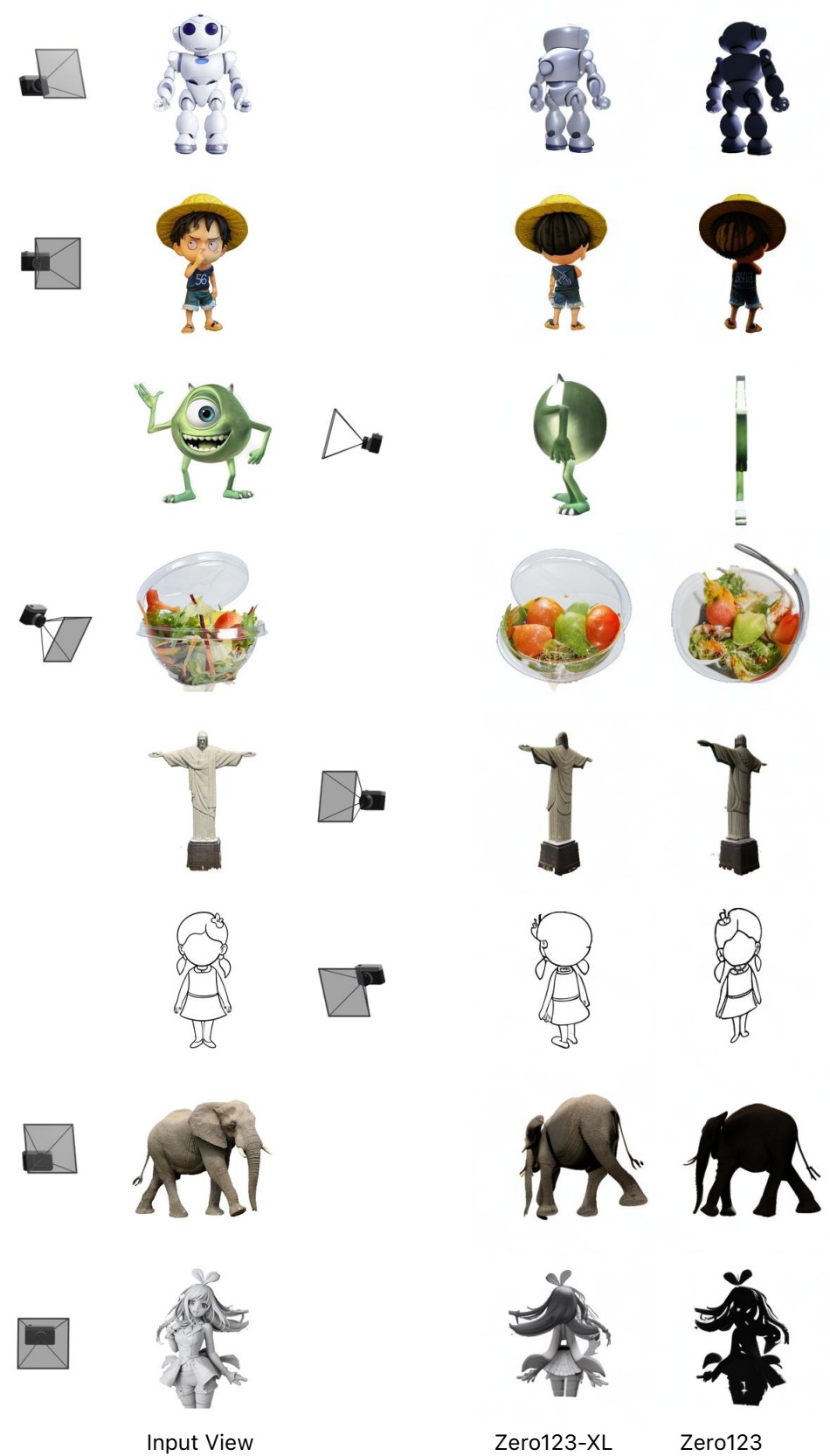

Input View            Zero123-XL      Zero123

Figure 17: Continuation of additional examples comparing Zero123-XL and Zero123.

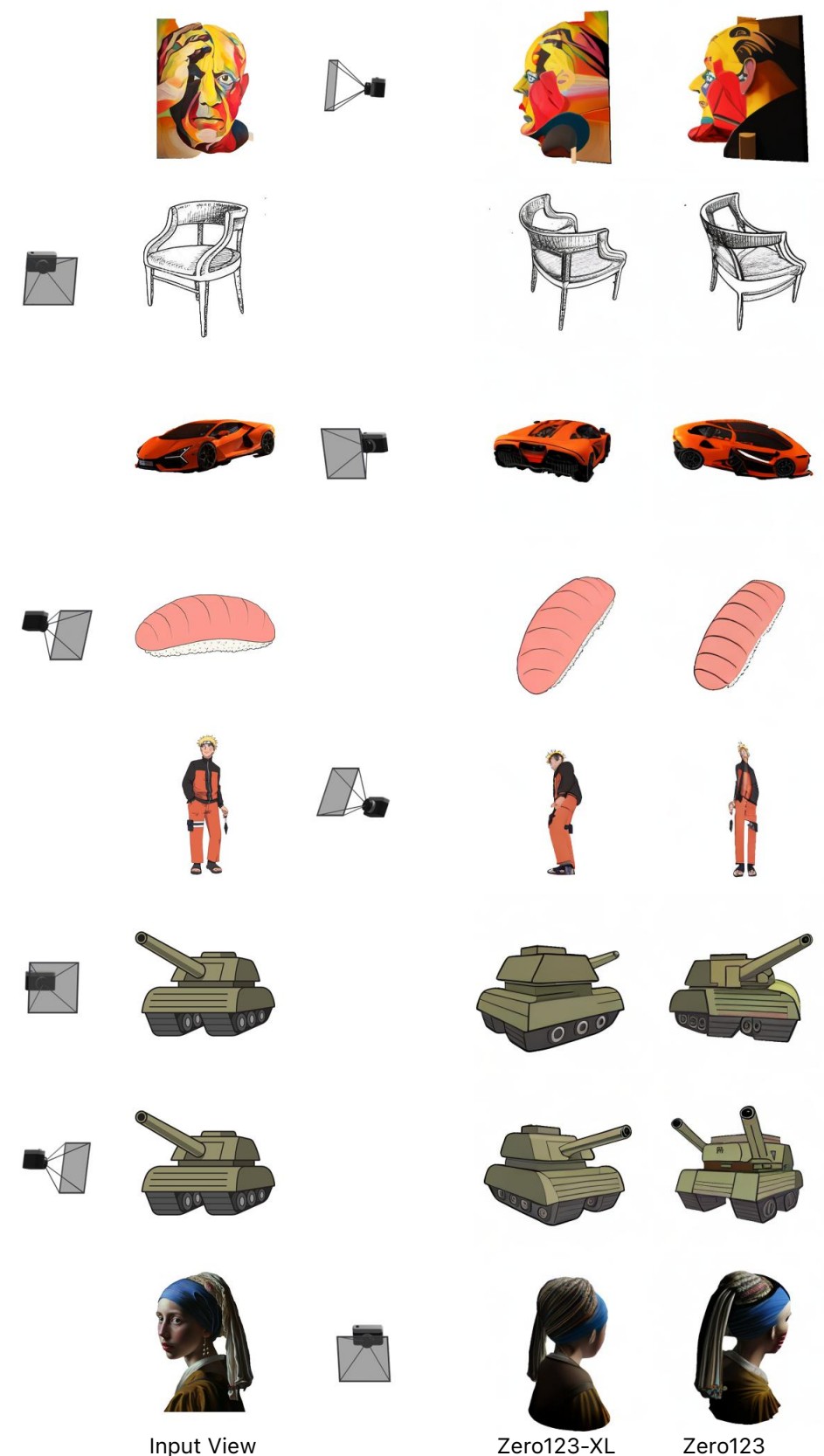

Input View        Zero123-XL     Zero123

Figure 18: Continuation of additional examples comparing Zero123-XL and Zero123.

# D Datasheet

This section provides a datasheet [71] for Objaverse-XL.

## D.1 Motivation

**For what purpose was the dataset created?** Was there a specific task in mind? Was there a specific gap that needed to be filled? Please provide a description.
The Objaverse-XL dataset was created to address the lack of high-quality, large-scale datasets for 3D vision tasks. This was due to challenges in acquiring such data and the associated complexities of 3D object generation and reconstruction, which were largely reliant on smaller, handcrafted datasets. The dataset was designed with the aim of advancing the field of 3D vision, allowing for the development and generalization improvement of models like Zero123 which work on tasks like novel view synthesis. The creation of Objaverse-XL essentially fills the gap in data availability for 3D vision tasks, particularly in light of increasing demand and interest in simulation, AR and VR technologies, and generative AI.

**Who created this dataset (e.g., which team, research group) and on behalf of which entity (e.g., company, institution, organization)?**
The dataset was created by researchers at the Allen Institute for AI and at the University of Washington, Seattle.

**What support was needed to make this dataset?** (e.g., who funded the creation of the dataset? If there is an associated grant, provide the name of the grantor and the grant name and number, or if it was supported by a company or government agency, give those details.)
Stability AI provided compute support and guidance for the main experiments in the paper. The Allen Institute for AI also provided compute support for collecting the dataset and performing rendering.

**Any other comments?**
No

## D.2 Composition

**What do the instances that comprise the dataset represent (e.g., documents, photos, people, countries)?** Are there multiple types of instances (e.g., movies, users, and ratings; people and interactions between them; nodes and edges)? Please provide a description.
Instances of the dataset comprise of 3D objects and their associated metadata.

**How many instances are there in total (of each type, if appropriate)?**
There are approximately 10.2 million rendered 3D files. About 56% come from GitHub, 35% come from Thingiverse, 8% come from Sketchfab, and less than 1% come from Polycam and the Smithsonian Institute. We also release additional links to indexed GitHub files that are not included in the count due to being removed by deduplication or not being easily importable into Blender.

**Does the dataset contain all possible instances or is it a sample (not necessarily random) of instances from a larger set?** If the dataset is a sample, then what is the larger set? Is the sample representative of the larger set (e.g., geographic coverage)? If so, please describe how this representativeness was validated/verified. If it is not representative of the larger set, please describe why not (e.g., to cover a more diverse range of instances, because instances were withheld or unavailable).
The dataset contains a sample of objects on GitHub, Sketchfab, Thingiverse, and Polycam, along with all the objects from the Smithsonian Institute.

**What data does each instance consist of?** "Raw" data (e.g., unprocessed text or images) or features? In either case, please provide a description.
Instances of the dataset vary based on source. For Polycam and Sketchfab objects, we release the full

3D objects along with associated metadata. For GitHub, Thingiverse, and Smithsonian objects, we release links to each of the files that can then be downloaded from the source, along with metadata such as license, poly count, vertex count, and other attributes discussed in Section 3.2.

**Is there a label or target associated with each instance?** If so, please provide a description.
No, just the 3D object and associated metadata. Labels and targets may be derived from the metadata, but vary based on the task.

**Is any information missing from individual instances?** If so, please provide a description, explaining why this information is missing (e.g., because it was unavailable). This does not include intentionally removed information, but might include, e.g., redacted text.
No, information is not missing.

**Are relationships between individual instances made explicit (e.g., users' movie ratings, social network links)?** If so, please describe how these relationships are made explicit.
Individual instances are treated as independent.

**Are there recommended data splits (e.g., training, development/validation, testing)?** If so, please provide a description of these splits, explaining the rationale behind them.
There are no recommended data splits across the entire dataset as splits vary based on task.

**Are there any errors, sources of noise, or redundancies in the dataset?** If so, please provide a description.
We deduplicated the objects by taking a sha256 of the file contents. There may still be near duplicates that exist, if the objects are slightly modified, which could potentially be filtered out, if desirable, using CLIP embeddings of the renders.

**Is the dataset self-contained, or does it link to or otherwise rely on external resources (e.g., websites, tweets, other datasets)?** If it links to or relies on external resources, a) are there guarantees that they will exist, and remain constant, over time; b) are there official archival versions of the complete dataset (i.e., including the external resources as they existed at the time the dataset was created); c) are there any restrictions (e.g., licenses, fees) associated with any of the external resources that might apply to a future user? Please provide descriptions of all external resources and any restrictions associated with them, as well as links or other access points, as appropriate.
There are no fees. The Smithsonian data is hosted on a governmental website, which we believe will be well supported over time. The Sketchfab and Polycam data will be available to download easily on our platform. For Thingiverse and GitHub, the platforms are relatively stable and we expect most of the content to remain in place. For Thingiverse, an API key must be provided to download the content from their API. For GitHub, the data can be easily cloned. Users must follow the license of the content with which the original files were distributed and the terms of service for each platform.

**Does the dataset contain data that might be considered confidential (e.g., data that is protected by legal privilege or by doctor-patient confidentiality, data that includes the content of individuals' non-public communications)?** If so, please provide a description.
While rare, it is possible that confidential data exists as part of the 3D objects on the platforms.

**Does the dataset contain data that, if viewed directly, might be offensive, insulting, threatening, or might otherwise cause anxiety?** If so, please describe why.
While rare, it is possible that data that is considered offensive, insulting, threatening, or might cause anxiety exists as part of the 3D objects on the platforms.

**Does the dataset relate to people?** If not, you may skip the remaining questions in this section.
People may be present in the dataset, but only make up a small part of it. Section discusses the results and analysis of running a face detector on renders of the objects. Most often, faces appear from dolls,

historic sculptures, and anthropomorphic animations. Moreover, even including such data, only about 2.5% captured faces.

**Does the dataset identify any subpopulations (e.g., by age, gender)?** If so, please describe how these subpopulations are identified and provide a description of their respective distributions within the dataset.
We do not identify people by subpopulation.

**Is it possible to identify individuals (i.e., one or more natural persons), either directly or indirectly (i.e., in combination with other data) from the dataset?** If so, please describe how.
If a scanned person is included in the dataset, it may be possible to visually identify them or identify them if their name is included as part of the metadata.

**Does the dataset contain data that might be considered sensitive in any way (e.g., data that reveals racial or ethnic origins, sexual orientations, religious beliefs, political opinions or union memberships, or locations; financial or health data; biometric or genetic data; forms of government identification, such as social security numbers; criminal history)?** If so, please provide a description.
While rare, it is possible that sensitive data may exist as part of the 3D objects on the platforms.

**Any other comments?**
No.

### D.3 Collection

**How was the data associated with each instance acquired?** Was the data directly observable (e.g., raw text, movie ratings), reported by subjects (e.g., survey responses), or indirectly inferred/derived from other data (e.g., part-of-speech tags, model-based guesses for age or language)? If data was reported by subjects or indirectly inferred/derived from other data, was the data validated/verified? If so, please describe how.
The data was directly observable and hosted on several platforms, including GitHub, Thingiverse, Sketchfab, Polycam, and the Smithsonian Institute.

**Over what timeframe was the data collected?** Does this timeframe match the creation timeframe of the data associated with the instances (e.g., recent crawl of old news articles)? If not, please describe the timeframe in which the data associated with the instances was created. Finally, list when the dataset was first published.
Sketchfab data was collected as part of Objaverse 1.0. The new data was collected in Q1 & Q2 of 2023.

**What mechanisms or procedures were used to collect the data (e.g., hardware apparatus or sensor, manual human curation, software program, software API)?** How were these mechanisms or procedures validated?
Python scripts were used to collect the data.

**What was the resource cost of collecting the data?** (e.g. what were the required computational resources, and the associated financial costs)
The cost of collecting the dataset was on the order of several thousand dollars, including costs to find, index, and download it. Resources included AWS CPU instances.

**If the dataset is a sample from a larger set, what was the sampling strategy (e.g., deterministic, probabilistic with specific sampling probabilities)?**
The dataset is filtered down based on licensing restrictions, duplicate content, and based on if the 3D object can successfully be imported into Blender.

**Who was involved in the data collection process (e.g., students, crowdworkers, contractors) and how were they compensated (e.g., how much were crowdworkers paid)?**
The data collection process was primarily performed by employed researchers at the Allen Institute for AI.

**Were any ethical review processes conducted (e.g., by an institutional review board)?** If so, please provide a description of these review processes, including the outcomes, as well as a link or other access point to any supporting documentation.
Institutional review boards were not involved in the collection of the dataset.

**Does the dataset relate to people?** If not, you may skip the remainder of the questions in this section.
People may be present in the dataset, but only make up a small part of it. Section D.2 discusses the results and analysis of running a face detector on renders of the objects. Most often, faces appear from dolls, historic sculptures, and anthropomorphic animations. Moreover, even including such data, only about 2.5% captured faces.

**Did you collect the data from the individuals in question directly, or obtain it via third parties or other sources (e.g., websites)?**
Data was collected from public facing platforms. On each of the platforms, users opted to make their data public.

**Were the individuals in question notified about the data collection?** If so, please describe (or show with screenshots or other information) how notice was provided, and provide a link or other access point to, or otherwise reproduce, the exact language of the notification itself.
Individuals were not notified about the collection of the dataset.

**Did the individuals in question consent to the collection and use of their data?** If so, please describe (or show with screenshots or other information) how consent was requested and provided, and provide a link or other access point to, or otherwise reproduce, the exact language to which the individuals consented.
Individuals were not notified about the collection of the dataset.

**If consent was obtained, were the consenting individuals provided with a mechanism to revoke their consent in the future or for certain uses?** If so, please provide a description, as well as a link or other access point to the mechanism (if appropriate)
Individuals were not notified about the collection of the dataset.

**Has an analysis of the potential impact of the dataset and its use on data subjects (e.g., a data protection impact analysis) been conducted?** If so, please provide a description of this analysis, including the outcomes, as well as a link or other access point to any supporting documentation.
An analysis has not been conducted.

**Any other comments?**
No.

## D.4   Processing / Cleaning / Labeling

**Was any preprocessing/cleaning/labeling of the data done(e.g., discretization or bucketing, tokenization, part-of-speech tagging, SIFT feature extraction, removal of instances, processing of missing values)?** If so, please provide a description. If not, you may skip the remainder of the questions in this section.
Preprocessing the data was performed by computing renders of the objects and performing

deduplication.

**Was the "raw" data saved in addition to the preprocessed/cleaned/labeled data (e.g., to support unanticipated future uses)?** If so, please provide a link or other access point to the "raw" data.
s The data that is downloaded contains the raw data and does not modify the individual files.

**Is the software used to preprocess/clean/label the instances available?** If so, please provide a link or other access point.
The software for cleaning the dataset and rendering will be made available.

**Any other comments?**
No.

## D.5 Uses

**Has the dataset been used for any tasks already?** If so, please provide a description.
Yes, please see Section 4 of the paper.

**Is there a repository that links to any or all papers or systems that use the dataset?** If so, please provide a link or other access point.
We recommend checking the Semantic Scholar page for the Objaverse-XL and Objaverse 1.0 papers to find up to date papers that use the dataset.

**What (other) tasks could the dataset be used for?**
A key use case of Objaverse-XL, and models such as Zero123-XL, is for making 3D content creation more accessible through generative modeling. We hope that by enabling a much wider audience to build content in 3D, it will enable many future applications in robotics simulation, AR/VR, games, education, architecture, manufacturing, healthcare, medicine, and biology, among many other industries. Beyond generation, Objaverse 1.0 has enabled many exciting research directions. We believe that the strong scaling trends exhibited in our paper suggest that Objaverse-XL will better empower similar tasks. In particular, we are excited to see Objaverse-XL be used for building on works that use Objaverse 1.0, such as in building 3D-LLMs [72], 3D retrieval systems [73, 74], representation learning on synthetic images [75], text-to-texture generation [76, 77], and robotic simulation and robustness [14], among others.

**Is there anything about the composition of the dataset or the way it was collected and preprocessed/cleaned/labeled that might impact future uses?** For example, is there anything that a future user might need to know to avoid uses that could result in unfair treatment of individuals or groups (e.g., stereotyping, quality of service issues) or other undesirable harms (e.g., financial harms, legal risks) If so, please provide a description. Is there anything a future user could do to mitigate these undesirable harms?
Users should follow the license of the individual objects distributed as part of this dataset.

**Are there tasks for which the dataset should not be used?** If so, please provide a description.
New tasks must make sure to follow the license of the dataset and the license of the individual objects distributed as part of the dataset.

**Any other comments?**
No.

### D.6 Distribution

**Will the dataset be distributed to third parties outside of the entity (e.g., company, institution, organization) on behalf of which the dataset was created?** If so, please provide a description.
Yes. The dataset will be made public.

**How will the dataset will be distributed (e.g., tarball on website, API, GitHub)?** Does the dataset have a digital object identifier (DOI)?
The dataset will be distributed through a Python API.

**When will the dataset be distributed?**
The dataset will be made publicly available towards the end of June, 2023.

**Will the dataset be distributed under a copyright or other intellectual property (IP) license, and/or under applicable terms of use (ToU)?** If so, please describe this license and/or ToU, and provide a link or other access point to, or otherwise reproduce, any relevant licensing terms or ToU, as well as any fees associated with these restrictions.
The dataset as a whole will be distributed under the ODC-By 1.0 license. The individual objects are subject to the licenses that they are released under, and users need to assess license questions based on downstream use.

**Have any third parties imposed IP-based or other restrictions on the data associated with the instances?** If so, please describe these restrictions, and provide a link or other access point to, or otherwise reproduce, any relevant licensing terms, as well as any fees associated with these restrictions.
The individual objects are subject to the licenses that they are released under, and users need to assess license questions based on downstream use.

**Do any export controls or other regulatory restrictions apply to the dataset or to individual instances?** If so, please describe these restrictions, and provide a link or other access point to, or otherwise reproduce, any supporting documentation.
No.

**Any other comments?**
No.

### D.7 Maintenance

**Who is supporting/hosting/maintaining the dataset?**
The dataset will be hosted on Hugging Face.

**How can the owner/curator/manager of the dataset be contacted (e.g., email address)?**
Please contact `mattd@allenai.org`.

**Is there an erratum?** If so, please provide a link or other access point.
No.

**Will the dataset be updated (e.g., to correct labeling errors, add new instances, delete instances)?** If so, please describe how often, by whom, and how updates will be communicated to users (e.g., mailing list, GitHub)?
The dataset is currently self contained without immediate plans for updates.

**If the dataset relates to people, are there applicable limits on the retention of the data associated with the instances (e.g., were individuals in question told that their data would be retained for a fixed period of time and then deleted)?** If so, please describe these limits and explain how they will be enforced.
People may contact us to add specific samples to a blacklist.

**Will older versions of the dataset continue to be supported/hosted/maintained?** If so, please describe how. If not, please describe how its obsolescence will be communicated to users.
Objaverse 1.0 will continue to be supported.

**If others want to extend/augment/build on/contribute to the dataset, is there a mechanism for them to do so?** If so, please provide a description. Will these contributions be validated/verified? If so, please describe how. If not, why not? Is there a process for communicating/distributing these contributions to other users? If so, please provide a description.
We encourage others to build and extend the dataset for different use cases and may highlight some of those use cases if applicable.

**Any other comments?**
No

## E   Aesthetic Annotations

We run LAION-Aesthetics V2 [12] on renders of the objects, which can be used for filtering a higher quality subset of the objects. We group the objects into 3 tiers, which are depicted in Table 4. Figure 19 shows examples of renders of objects placed on the different tiers.

| Category | Description | Aesthetic Score Cutoff | Percentage of Dataset |
|---|---|---|---|
| T1 | Highest aesthetic ranked objects | Greater than 4.5 | 14.2% |
| T2 | Medium aesthetic ranked objects | Between 4 and 4.5 | 69.2% |
| T3 | Low aesthetic ranked objects | Less than 4 | 16.6% |

Table 4: LAION-Aesthetics V2 categorization for renders of Objaverse-XL objects.

T1 Aesthetic Tier

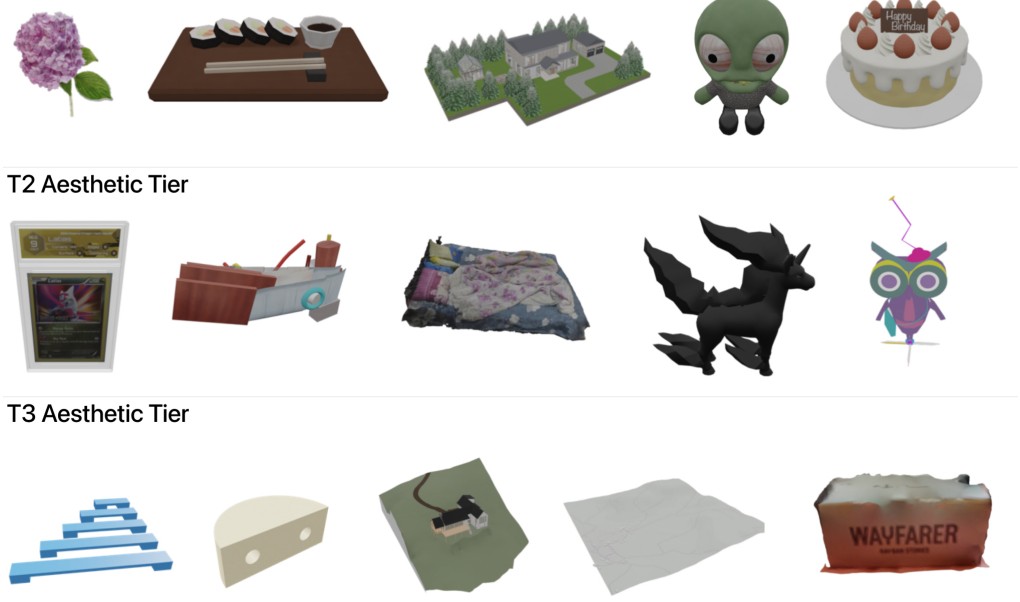

T2 Aesthetic Tier

T3 Aesthetic Tier

Figure 19: Random samples of renders showing LAION-Aesthetic V2 annotations across different tiers. Empirically, T1 tends to have the highest quality objects, followed by T2 and then T3.

# F    Zero123-XL Human Evaluation

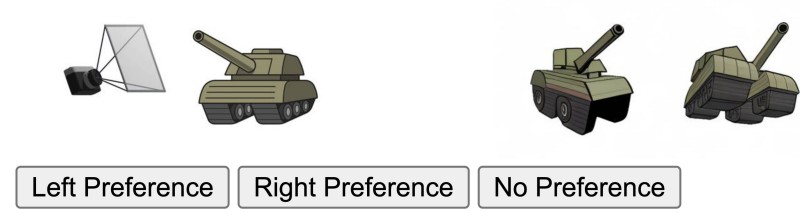

Figure 20: The interface used for human evaluation.

To study the significance of the Zero123-XL results, we performed human evaluation to determine if users preferred the results of Zero123-XL compared to Zero123. Figure 20 shows the interface for the evaluation experiments. In particular, the left side shows the input image and the camera transformation, and the right side shows the novel view synthesis results from both Zero123-XL and Zero123 (in a randomized order). If the user has a preference between the results, the user is then asked to select which novel view synthesis result they prefer, the one on the left or right. The human evaluation was performed by employees at the Allen Institute for AI.

Our study found that users preferred the novel view synthesis results from Zero123-XL 62.19% of the time compared to the baseline Zero123 model. The z-score comparing the proportion of votes between the two models is approximately 4.24, which is much larger than the typical critical value of 1.96 for a two-tailed test with a significance level of $\alpha = 0.05$. Thus, we would reject the null hypothesis, and suggest that there is a statistically significant difference between the preferences from the outputs of the models in terms of their voting proportions.

## G    Evaluation Statistical Significance

In this section, we provide statistical significance tests for the Zero123-XL and PixelNeRF evaluations. We find that on all experiments, the models trained with Objaverse-XL perform statistically significantly better than the baselines that do not use Objaverse-XL.

We use the paired $t$-test, which tests the statistical significance of the two models when evaluated on the same data points. The null hypothesis for the paired $t$-test is that the differences between the paired observations come from a distribution with a mean equal to zero. If the $t$-test rejects the null hypothesis, this suggests that there is a statistically significant difference between the means of the two groups. The $t$-statistic is a measure of the difference between the two sample means in units of the standard error, and it allows us to determine how extreme the observed sample difference is, assuming the null hypothesis is true. The associated $p$-value quantifies the probability of observing a $t$-statistic as extreme as, or more extreme than, the one computed from the data, under the null hypothesis. In essence, a smaller p-value indicates stronger evidence against the null hypothesis. We use the standard threshold that a p-value below $0.05$ is considered evidence of a statistically significant effect, meaning there's a less than 5% probability of observing the given result if the null hypothesis were true.

| | PSNR | | SSIM | | LPIPS | |
| Benchmark | $t$-statistic ($\uparrow$) | $p$-value | $t$-statistic ($\uparrow$) | $p$-value | $t$-statistic ($\downarrow$) | $p$-value |
|---|---|---|---|---|---|---|
| ShapeNet | 3.27 | 0.0028 | 2.71 | 0.0113 | $-4.42$ | $2.81 \times 10^{-5}$ |
| DTU | 4.81 | $6.20 \times 10^{-6}$ | 4.56 | $8.5 \times 10^{-5}$ | $-4.16$ | $2.58 \times 10^{-4}$ |

Table 5: The paired $t$-test statistical significance results for PixelNeRF evaluated on ShapeNet and DTU that compare models trained with Objaverse-XL to the baseline trained without it.

| PSNR | | SSIM | | LPIPS | |
| $t$-statistic ($\uparrow$) | $p$-value | $t$-statistic ($\uparrow$) | $p$-value | $t$-statistic ($\downarrow$) | $p$-value |
|---|---|---|---|---|---|
| 6.02 | $3.79 \times 10^{-8}$ | 3.334 | 0.0013 | $-11.10$ | $1.77 \times 10^{-18}$ |

Table 6: The paired $t$-test statistical significance results when comparing Zero123-XL to Zero123 on 0-shot evaluation with Google Scanned Objects.

For the PixelNeRF experiments, we compare the results of a model pre-trained on 2 million Objaverse-XL objects compared to one initialized from scratch on the standard novel view synthesis benchmarks for ShapeNet and DTU. For the Zero123-XL experiment, we compare the results of Zero123 and Zero123-XL in a 0-shot setting on Google Scanned Objects. Table 5 shows our results for the PixelNeRF experiments and Table 6 shows the results for the Zero123-XL experiments. In particular, we find that all of our evaluations are statistically significant, with a $p$-value of less than $0.05$.

## H    Image-to-3D with Zero123-XL

After training Zero123-XL to perform an image-to-image transformation under a new camera transformation, we can then use the model to perform single-view 3D reconstruction. The process works by swapping out the text-to-image model in Dreamfusion [47] with Zero123-XL, to train a Neural Radiance Field (NeRF) [38] model with diffusion guidance to generate coherent views from each side of the object, given a single starting view, and estimated transformations produced by Zero123-XL. More details on the process of obtaining going from image-to-3D are available in the Zero123 paper [15].

Figure 21 shows examples of performing image-to-3D with Zero123-XL. Moreover, we can also easily perform text-to-3D, by first using a text-to-image model to obtain an initial image, and then using the image-to-3D model to obtain the 3D object. Figure 21 also shows an example of text-to-3D using text-to-image-to-3D with Stable Diffusion XL (SDXL) [78]. More details and the implementation are available at https://github.com/threestudio-project/threestudio#zero-1-to-3-.

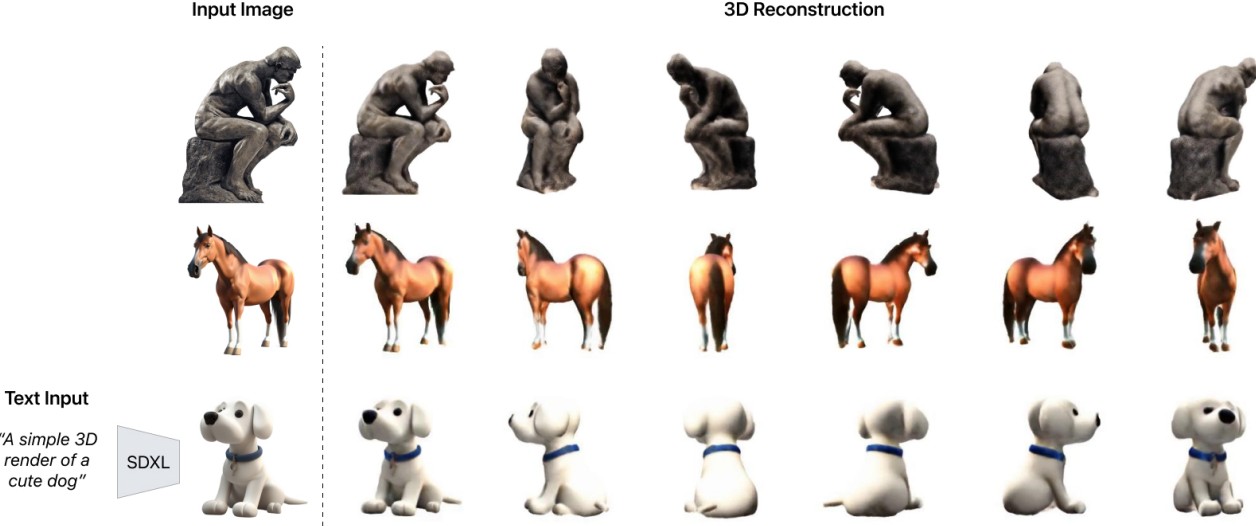

Figure 21: Image-to-3D results using Zero123-XL. We also show text-to-3D by first performing text-to-image and then performing image-to-3D.

# I Failure Cases

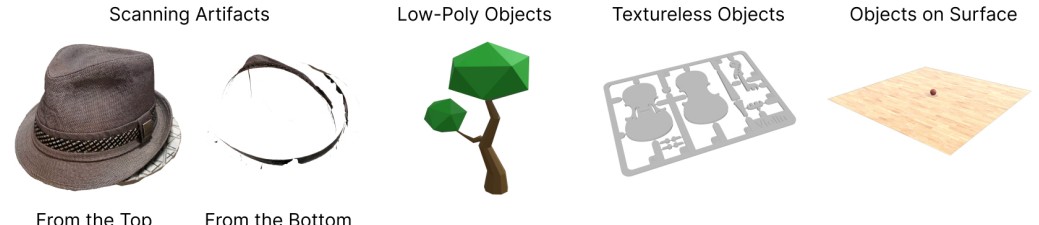

Figure 22: Possible sources of undesirable objects include those with scanning artifacts, low-poly objects, textureless objects, and objects that appear on a surface.

Figure 22 shows examples of possible sources of undesirable objects, depending on the downstream task. For scanned objects, many of the objects have holes in areas that were not scanned (which often includes the bottom side of the object). Additionally, objects may have a low polygon count, be textureless (e.g., from Thingiverse), or be on a large surface that was used for rendering. Being on a surface is undesirable because objects are normalized to the size of a unit cube when rendered, making the object of interest extremely tiny.

Additionally, Figure 23 shows some examples of failure cases from the Zero123-XL model. Similar to text-to-image models, it occasionally struggles with counting and the relative positions of different objects. Some artifacts such as the bottom of objects being unclosed may also be observed, which likely comes from the scanning artifacts in the dataset. Moreover, the model sometimes smudges details, which are noticeable when in faces.

# J Further Dataset Analysis

In this section, we provide several more visualizations to analyze the Objaverse-XL object distribution. Here, the analysis is performed on a uniform sample of 10K objects from the dataset.

In Figure 24, we show the density plots for 3 more attributes captured in the metadata during rendering, including the densities for the object counts (the number of items in each object's hierarchy), the number of materials associated with each object, and the size of the files (in megabytes (MB)). As we can see, each object's hierarchy often has a considerable number of objects associated with it, many

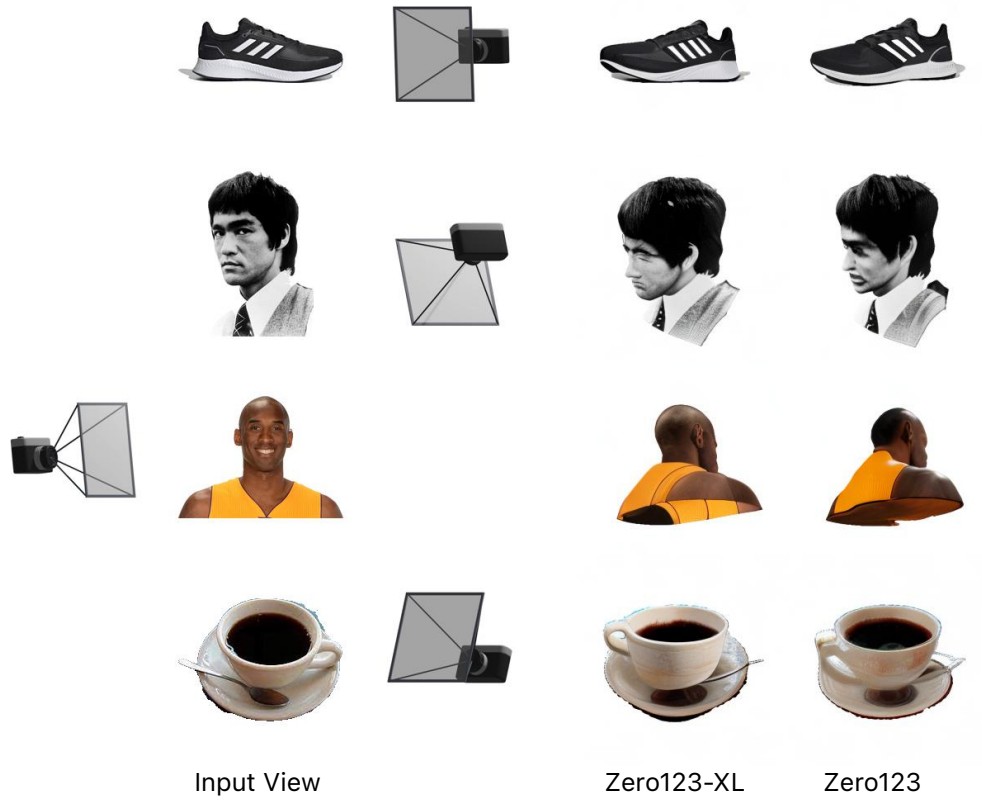

Input View          Zero123-XL      Zero123

Figure 23: Failure cases from Zero123-XL may include counting artifacts, difficulties with smudging faces, occasional unclosed objects on the bottom, and relative positional issues when working with multiple objects.

objects have materials, and most object files are less than 10 MB, with each of these distributions following a long-tail.

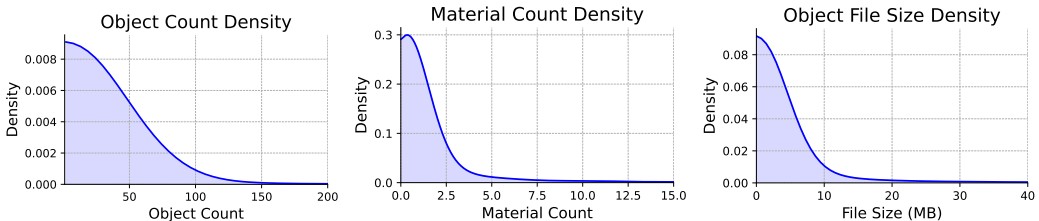

Figure 24: The density plots of several attributes of the objects, including the object count (the number of items each object's hierarchy), the number of materials associated with the object, and the size of the files (in MB).

In Figure 25, we show the scatter plot matrix of the different aspect ratios of the objects compared to each other. It is plotted on a log scale for visualization purposes. Note that $z$ is the upward facing direction. One interesting finding can be observed from the linear correlation between $\log(y/z)$ and $\log(x/z)$, which suggests a proportional relationship between the width and depth of the 3D models relative to their height.

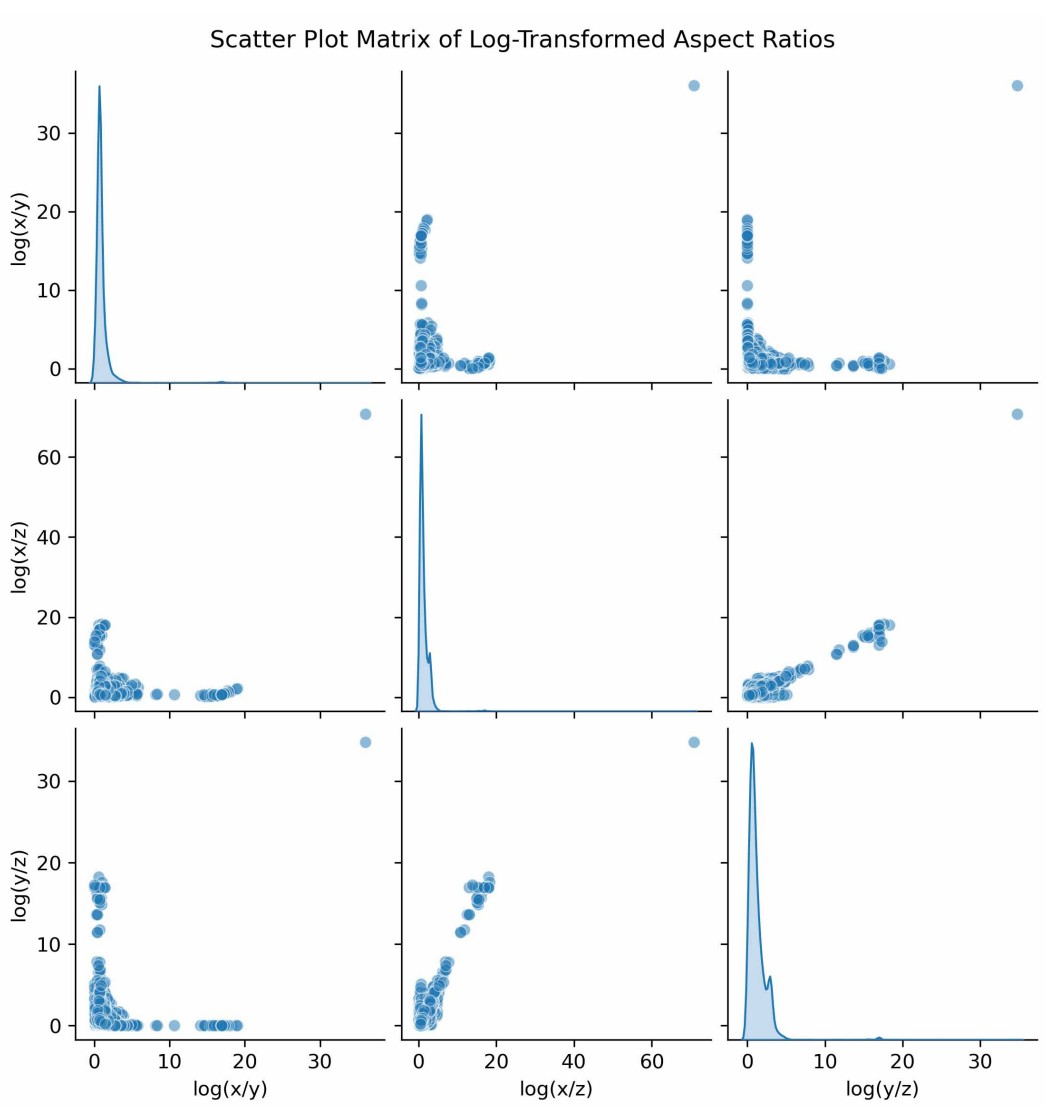

Figure 25: The scatter plot matrix of the aspect ratios of the objects. Note that $z$ is the upward facing direction.