# OpenReview forum: "Objaverse-XL: A Universe of 10M+ 3D Objects"
_NeurIPS.cc/2023/Track/Datasets_and_Benchmarks — NeurIPS 2023 Datasets and Benchmarks Poster_

### Official Review · Reviewer_toL1 · 2023-07-06
**Objaverse-XL makes a huge impact on the field of 3D vision tasks**

**Rating:** 7
**Confidence:** 4

**Strengths:**

+Scale and Diversity: Objaverse-XL is a dataset of over 10 million 3D objects, representing the largest scale and diversity in the realm of 3D datasets. This extensive collection opens up significant new possibilities for 3D vision research.

+Comprehensive Compilation: The dataset comprises deduplicated 3D objects sourced from various channels, including manually designed objects, photogrammetry scans of landmarks and everyday items, as well as professional scans of historic and antique artifacts. This comprehensive compilation ensures a wide range of object types and characteristics.

+Improved Results: Experiments conducted with Objaverse-XL demonstrate significant improvements achieved through its scale. Training the Zero123 model on novel view synthesis using over 100 million multi-view rendered images yields strong zero-shot generalization abilities. This suggests that the dataset enables more accurate and effective training of 3D vision models.

+Potential for Innovation: By releasing Objaverse-XL, the authors aim to enable further innovations in the field of 3D vision at scale. The dataset provides researchers with a valuable resource to advance their understanding and develop novel approaches in 3D vision tasks.

**Additional Feedback:**

I would suggest the author to design some interface to allow the public to contribute to the dataset also.

**Clarity:**

The structure and organization of the paper are reasonable, and the flow of the content is smooth. No obvious spelling errors were found.

**Correctness:**

The relevant statements in the paper are appropriate. The construction of the dataset is reasonable, and the sources of the data are provided. Additionally, the code for the benchmark has been made openly available. The benchmark evaluation method is suitable and can be easily reproduced.

**Documentation:**

The paper provides a detailed description of the dataset creation process, and there are no ethical concerns regarding its production. The authors have made the dataset openly available on GitHub under the CC BY 4.0 license. The benchmark's environment setup, dataset structure partitioning, and reproducibility methods are also thoroughly described on GitHub.

**Ethics:**

I believe that the submitted content does not raise any ethical concerns

**Limitations:**

Yes

**Opportunities For Improvement:**

It would make the dataset even better if some handy APIs could be provided and some interface is provided to allow the public to contribute to the dataset also.

**Relation To Prior Work:**

This seems to be a enhanced version of previous dataset "Objaverse: A Universe of Annotated 3D Object"

**Summary And Contributions:**

This work addresses the limited progress in 3D vision tasks compared to natural language processing and 2D vision models. The main challenge is the scarcity of high-quality 3D data. To overcome this, the authors introduce Objaverse-XL, a dataset containing over 10 million 3D objects. The dataset is compiled from various sources, including manually designed objects, photogrammetry scans of landmarks and everyday items, and professional scans of historical artifacts. Objaverse-XL represents the largest and most diverse collection of 3D data, unlocking new possibilities for 3D vision research. The experiments conducted using Objaverse-XL demonstrate significant improvements made possible by its scale. In particular, training a model called Zero123 on novel view synthesis using over 100 million multi-view rendered images enables strong zero-shot generalization capabilities. The release of Objaverse-XL is expected to facilitate further innovations in the field of 3D vision on a large scale.

---

> ### Author Response · Authors · 2023-08-23
> **Author Response**
>
> We thank the reviewer for their review. We are glad they found the dataset “opens up significant new possibilities for 3D vision research,” “enables more accurate and effective training of 3D vision models,” and that “the flow of the content is smooth.”
>
> > Is there a way for the public able to contribute to the dataset?
> >
>
> Yes, this is possible! On Hugging Face, people can make pull requests ([here](https://huggingface.co/datasets/allenai/objaverse-xl/discussions?status=open&type=pull_request)), which will allow others to possibly contribute new 3D objects or updates to the API!
>
> Moreover, with Objaverse 1.0, we have already seen the community distribute and start to work on different annotations, such as captions (e.g., from [Cap3D](https://arxiv.org/abs/2306.07279)), point clouds (e.g., from [ULIP-2](https://github.com/salesforce/ULIP#news)), and rendered images (e.g., from [OpenShape](https://github.com/Colin97/OpenShape_code)). We are hopeful that the community will continue in this direction with Objaverse-XL!
>
> > This seems to be a enhanced version of previous dataset "Objaverse: A Universe of Annotated 3D Object"
> >
>
> That’s correct. Objaverse 1.0 included 800K objects from Sketchfab. To further push the limits of scaling in the realm of 3D research, we built Objaverse-XL to be a much larger dataset, consisting of 10M+ 3D objects from various sources across the web. We show strong scaling trends when going from training on Objaverse 1.0 to training on Objaverse-XL, and with Zero123-XL, even display strong 3D generalization results when evaluating novel view synthesis on a long tail distribution of concepts, such as on sketches, cartoons, and photos of people.
>
> Please let us know if you have any additional questions!
>
> Finally, if you believe this paper should be highlighted at the conference, could you please consider raising your score to reflect that?

---

> ### Author Response · Authors · 2023-08-28
> **Rebuttal Reminder**
>
> Hi Reviewer toL1,
>
> Just a quick reminder that the rebuttal deadline is **tomorrow**, August 29th at **01:00 PM PDT**. Please reach out if you need more clarification or if you have further questions regarding the paper!
>
> Best regards.

---

### Official Review · Reviewer_QWh5 · 2023-07-16
**Objaverse-XL - A large scale 3D object dataset**

**Rating:** 6
**Confidence:** 4

**Strengths:**

+ The scale of the proposed dataset is significantly larger than existing 3D object dataset
+ The paper presents qualitative and quantitative results showing the effects of data scale on novel view synthesis
+ The authors adopt data validation and deduplication approaches to ensure the usability of the dataset

**Additional Feedback:**

N/A

**Clarity:**

- Since the 3D objects are collected from multiple sources, they can be quite different in terms of quality, file formats and resolution. It is necessary to share more preprocessing details for each data source. For example, how the color is randomized for untextured objects from thingiverse?
- The license is not clear: the objaverse-XL dataset is released under ODC-by license. However, it seems individual objects are subject to their original licenses. Can we ensure that the user can retrieve the license of each individual object from the metadata? Otherwise it makes it very difficult for users to comply with the license

**Correctness:**

- One of the major claim of the paper is that dataset scale is crucial for downstream applications. Thus it is necessary to extend the original objaverse dataset to sth that is of large scale and probably of lower quality. Table 2 shows that finetuning the model on a 1 million high-quality subset of objaverse-XL outperforms the base pretrained model. What is the model's performance if pretrained on the high quality subset?
- For the novel view synthesis experiments, I was not able to find the information regarding the held-out subset used for evaluation. In addition, why not evaluate on an existing 3D object dataset?

**Documentation:**

- In the appendix, the authors state that the dataset will be shared with reviewers. However, so far I was not able to access the dataset.

**Limitations:**

- It seems the metadata do not include object category information. How are users supposed to retrieve the objects given a specific use case?
- Not all the objects in the dataset are of high quality. E.g. objects in Thingiverse have no texture and the resolution is typically quite low.

**Opportunities For Improvement:**

- The current experiment only includes the novel view synthesis use case. It would be great if the authors can show how the dataset can be used for other downstream applications

**Relation To Prior Work:**

The authors are recommended to cite the following work:
Song, S., Yu, F., Zeng, A., Chang, A. X., Savva, M., & Funkhouser, T. (2017). Semantic scene completion from a single depth image. In Proceedings of the IEEE conference on computer vision and pattern recognition (pp. 1746-1754).
Chang, A., Dai, A., Funkhouser, T., Halber, M., Niessner, M., Savva, M., ... & Zhang, Y. (2017). Matterport3d: Learning from rgb-d data in indoor environments. arXiv preprint arXiv:1709.06158.

**Summary And Contributions:**

This paper proposes objaverse-XL, a dataset containing more than 10 million 3D objects. The work extends the objaverse 1.0, a 800k 3D object dataset, by collecting objects from mulitple data sources. Experiment results show novel view synthesis models, evaluated on a held-out subset, demonstrate better performance when trained with a larger scale of data.

---

> ### Author Response · Authors · 2023-08-23
> **Author Response**
>
> We thank the reviewer for their review. We are glad they highlighted that our “dataset is significantly larger than existing 3D object [datasets]”, that our experiments show the “effects of data scale on novel view synthesis,” and that we “ensure the usability of the dataset.”
>
> > The current experiment only includes the novel view synthesis use case. It would be great if the authors can show how the dataset can be used for other downstream applications.
> >
>
> Thanks for the suggestion! We have added Section H to the appendix, which shows how we can use Zero123-XL to perform single image → 3D generation and text → 3D generation. We’ve added our implementation into [Threestudio](https://github.com/threestudio-project/threestudio#zero-1-to-3-), a popular framework for 3D content generation, to allow others to generate models as well.
>
> Additionally, while we focus on the dataset in this paper, and provide considerable generation experiments, study scaling trends, and build Zero123-XL to serve as a foundation model for 3D research, we agree that this dataset has a significant amount of value beyond generation. Since the release of Objaverse 1.0, it has enabled a tremendous amount of exciting research, and we expect each of these applications to similarly be able to benefit from Objaverse-XL. In particular, we are excited to see Objaverse-XL be used for building on works that use Objaverse 1.0, such as in building 3D-LLMs [[1](https://arxiv.org/abs/2307.12981)], 3D retrieval systems [[2](https://arxiv.org/abs/2305.08275), [3](https://arxiv.org/abs/2305.10764)], representation learning on synthetic images [[4](https://arxiv.org/abs/2308.03977)], text-to-texture generation [[5](https://arxiv.org/abs/2306.06212), [6](https://arxiv.org/abs/2303.11396)], and robotic simulation and robustness [[7](https://arxiv.org/abs/2212.08051)], among other applications in AR/VR, multimodal learning, and generation. We believe that the strong scaling trends exhibited in our paper suggest that Objaverse-XL will similarly be useful for many other tasks.
>
> > How are users supposed to retrieve the objects given a specific use case without using categories?
> >
>
> Category annotations are often not practical for internet-scale datasets, such as LAION-5B or web-text crawled from Common Crawl, as most items cannot be cleanly mapped to a set of categories that is universally useful for retrieval or filtering.
>
> Instead, similarity search with filtering is often what is most practically useful. Here, one can leverage embeddings of the objects (e.g., CLIP embeddings of rendered images of the objects), and metadata for filtering, to retrieve objects based on a specific use case. In this work, we already demonstrate two specific use cases that such an approach works well on: (1) finding our alignment finetuning subset of objects and (2) finding objects that are aesthetically pleasing. The metadata can be used to filter objects by popularity on the source website (e.g., likes on Sketchfab) or to have a minimum polygon count cutoff threshold, for example.
>
> There is also a line of work exploring retrieval systems for 3D objects, such as [OpenShape](https://huggingface.co/spaces/OpenShape/openshape-demo) and [ULIP-2](https://arxiv.org/abs/2305.08275), which leverage Objaverse 1.0. In particular, they support retrieving objects that are most similar to either a text description, input image, or a 3D point cloud, and we believe such approaches could neatly be extended to work with Objaverse-XL.
>
> > Dealing with dataset noise and low quality objects.
> >
>
> We take the position that many forms of noise are task-dependent and we let users select the objects they find useful. For example, filtering out low polygon count objects may be desirable in a photorealistic simulator, but they may be useful for mesh generation to learn a shape prior. Moreover, internet-scale datasets, such as LAION-5B and web-text from Common Crawl are notoriously noisy. Nevertheless, such noisy internet-scale datasets have been used to power many of the most significant recent advances in AI.

---

> > ### Author Response · Authors · 2023-08-23
> > **Author Response (Part 2)**
> >
> > > What is the model's performance if pre-trained on the high quality subset?
> > >
> >
> > We expect the model’s performance when pre-trained on the high quality subset to be similar to the model pre-trained on Objaverse 1.0; that is, pretty good, but does not capture the long tail (e.g., of sketches, people, animals, cartoons, and other “in-the-wild” images) as well as the model that trains on the entire Objaverse-XL dataset. We believe there is analogous observation with LLMs, where models trained with a small amount of expert data superficially follow the patterns observed in the data at first glance, but end up having many more flaws and are much more brittle [[1](https://arxiv.org/abs/2305.15717)].
> >
> > In this paper, we decided to pre-train Zero123-XL on the entire dataset, as we wanted to expose it to as many shapes as possible and build a strong shape prior into it, before then shallowly performing alignment fine-tuning with more traditional objects. Since our goal is for Zero123-XL to serve as the foundation for future models, we heavily emphasized wanting it to work well “in-the-wild” to enable a wide range of use cases.
> >
> > To ablate the Zero123-XL pre-training experiments, it would require a considerable amount of GPU time (on the order of 25K GPU hours), and we did not have the ability to run it during the rebuttal phase. Nevertheless, it is currently an exciting research direction to determine subsets of a dataset that can reach similar performance to the whole dataset, as training models on the subset would be more efficient than training it on the entire dataset (e.g., see [DataComp](https://arxiv.org/abs/2304.14108)). We hope to explore this question further in the future!
> >
> > > It is necessary to share more preprocessing details for each data source. For example, how is the color is randomized for untextured objects from Thingiverse?
> > >
> >
> > The preprocessing and rendering scripts will all be made open source and available to the community. We spent a considerable amount of time dealing with different file format compatibility constraints and we hope our preprocessing code makes it seamless to work with objects of many 3D file formats and from many different sources.
> >
> > For the color randomization performed on Thingiverse objects, the entire color of the objects are uniformly randomized, where the RGB values are each independently sampled uniformly between 0-100%.
> >
> > > When evaluating PixelNeRF on Objaverse-XL, what are the held-out evaluation dataset details?
> > >
> >
> > We have added these details to the appendix in the PixelNeRF implementation details section. In particular, for evaluation, we randomly sample 100,000 objects from Objaverse-XL, which consist of 1.2 million rendered images that are disjoint from the training set.
> >
> > > In addition, why not evaluate on an existing 3D object dataset?
> > >
> >
> > We do evaluate on existing 3D object datasets. For the Zero123-XL experiment, it is evaluated in a 0-shot setting on Google Scanned Objects (see Table 2). For the PixelNeRF experiments, it is additionally evaluated on both ShapeNet and DTU (see Table 3).
> >
> > > The license is not clear. Objaverse-XL is released under ODC-by license. However, it seems individual objects are subject to their original licenses.
> > >
> >
> > We believe the license is clear, and have consulted our legal counsel when drafting the license. To clarify the terms of the license, the use of Objaverse-XL as a whole is licensed under the ODC-by license. So, if you use any part of Objaverse-XL, you are governed by the ODC-by license. Then, if you use particular objects that are part of Objaverse-XL, you’re also governed by the licenses of those individual objects as they were released by the creators. These terms in the license are the same as what is specified in Objaverse 1.0.
> >
> > > Can we ensure that the user can retrieve the license of each individual object from the metadata?
> > >
> >
> > Yes. The metadata will provide the license of each individual object.
> >
> > > Pointers to cite SUNCG and Matterport3D related work.
> > >
> >
> > Thank you for the suggestion! We have added Matterport3D to the related works, and also swapped SUNCG with Habitat-Matterport 3D a similarly related scene dataset, as SUNCG is no longer available for download.
> >
> > > What is the link for the dataset?
> > >
> >
> > The dataset is available on Hugging Face at [https://huggingface.co/datasets/allenai/objaverse-xl](https://huggingface.co/datasets/allenai/objaverse-xl).
> >
> > We have also provided a download API and tutorial to get started working with it!
> >
> > Please let us know if you have any additional questions! We are happy to answer them and look forward to our discussion with you in the rebuttal phase.

---

> > ### Comment · Reviewer_QWh5 · 2023-08-29
> >
> > I'd like to thank the authors for their comprehensive response to my and other reviews' comments. The updated manuscript greatly improves the clarity over the original manuscript and provide essential information on data collection and experiment details. However, some original concerns remain:
> >
> > - Regarding the additional downstream applications, the revised manuscript added some qualitative results for text to 3D and image to 3D tasks. Nevertheless, it is still not very clear how much all the other applications mentioned (e.g. 3D-LLMs, 3D retrieval systems, representation learning on synthetic images, text-to-texture generation, and robotic simulation and robustness) can benefit from Objaverse-XL on top of the existing Objaverse 1.0.
> >
> > - The current results in Table 2 only show that finetuning the model on a smaller but higher quality dataset performs better than pretraining the model on a big dataset alone. While pretraining the model on the entire Objaverse-XL dataset and finetuning it on a high quality subset makes sense, it would be great to have some ablation study showing how the quality and scale of the dataset separately contribute to the model performance (e.g. finetune the pretrained model with a randomly sampled subset).

---

> > > ### Author Response · Authors · 2023-08-29
> > > **Response to Reviewer QWh5**
> > >
> > > We thank the reviewer for their response and are glad to hear that our initial responses “greatly improves the clarity over the original manuscript.” Below, we address each of the remaining comments.
> > >
> > > > It is still not very clear how much all the other applications mentioned (e.g. 3D-LLMs, 3D retrieval systems, representation learning on synthetic images, text-to-texture generation, and robotic simulation and robustness) can benefit from Objaverse-XL on top of the existing Objaverse 1.0.
> > > >
> > >
> > > Across AI, scaling datasets have enabled many recent breakthroughs in the generalization and robustness of models, including with CLIP (see analysis on the dataset’s contributions [here](https://arxiv.org/abs/2205.01397)), Flamingo, AlphaFold, MuZero Whisper, and LLMs, including GPT-4, LLAMA, Claude, and BERT, among others. Focusing on text-to-image models, the benefits of large-scale data have even been demonstrated across several different model families, including diffusion models such as Stable Diffusion, DALL-E 2, and Imagen; autoregressive models such as Muse and Parti; and GANs such as GigaGAN / StyleGAN-T. The benefits of large-scale data have been extensively demonstrated in these applications, as have the use of scaling trends to show the benefit of large-scale data (see: [Chinchilla](https://arxiv.org/abs/2203.15556), [Neural Scaling Laws](https://arxiv.org/abs/2001.08361), [CLIP](https://arxiv.org/abs/2103.00020), [Whisper](https://arxiv.org/abs/2212.04356)).
> > >
> > > For training Zero123-XL, which was done with 25k GPU hours, it costs around $100K for training [1]. As we are academics, even running an experiment at this scale was a great undertaking. So, we chose to focus on the application we thought had the most promise (training Zero123-XL for novel view synthesis) to demonstrate the potential of the dataset, and to show strong scaling trends that indicate the benefits of scale in 3D. We believe our 0-shot generalization results from Zero123-XL, our demonstrated scaling trends, and the demonstrated benefits of scaling data across AI indicate that the dataset is useful for many other tasks in 3D, and leave that exploration to future research.
> > >
> > > [1] The calculation comes from the AWS p4d.24xlarge instances, which have 8 A100s, costing [$32.7726 / hour](https://aws.amazon.com/ec2/pricing/on-demand/). For 25k GPU hours, that’s (32.7726 / 8) * 25000 = $102,414.
> > >
> > > > The current results in Table 2 only show that finetuning the model on a smaller but higher quality dataset performs better than pretraining the model on a big dataset alone. While pretraining the model on the entire Objaverse-XL dataset and finetuning it on a high quality subset makes sense, it would be great to have some ablation study showing how the quality and scale of the dataset separately contribute to the model performance (e.g. finetune the pretrained model with a randomly sampled subset).
> > > >
> > >
> > > Thanks for the suggestion, and we agree that this would be an interesting ablation experiment! From our intuition, we believe that training the pre-trained model longer, specifically fine-tuned on a similarly sized randomly sampled subset of the data for a duration similar to that from alignment fine-tuning, would result in a model that is similar, but not strikingly better than the current pre-trained model. That is, we believe the model will still be quite good at predicting shapes and novel views of objects, but the predictions would not look as realistic as the model fine-tuned on the alignment fine-tuning subset of the dataset, which is meant to have a lot more coverage of real world objects. What we’d expect is that objects would still be geometrically consistent, but there would be a much wider variety of sampled appearances of novel views, such as furniture appearing fantastical or often unnaturally stylized.
> > >
> > > We believe that our observation is also consistent with the 2 stage training approach happening with LLMs, where models are first pre-trained on a large corpus of text, then fine-tuned on a high-quality subset of data. Here, the large-scale datasets greatly improve the robustness of the models, and the high-quality fine-tuning makes the models more usable to human users.
> > >
> > > Currently, we do not have the resources to run this ablation right now, as conducting a similar fine-tuning ablation requires ~$15K worth of compute. Nevertheless, we will try our best to put an ablation that fine-tunes the pre-trained model with a randomly sampled subset in the camera-ready version!
> > >
> > > We hope that the rebuttal clarifies the questions raised by the reviewer. We would be very happy to discuss any further questions about the work, and would really appreciate an appropriate increase in the score if the reviewer’s concerns are adequately addressed to facilitate acceptance of the paper.

---

> > > > ### Comment · Reviewer_QWh5 · 2023-08-31
> > > >
> > > > I'd like to thank the authors again for the discussions. Overall I'm satisfied with the revisions as it provides additional analysis, necessary implementation details and valuable experiment results to demonstrate the usability of the dataset. I would like to increase my score to 6 to reflect these improvements and I look forward to the ablation study on the dataset quality!

---

> ### Author Response · Authors · 2023-08-28
> **Rebuttal Reminder**
>
> Hi Reviewer QWh5,
>
> Just a quick reminder that the rebuttal deadline is **tomorrow**, August 29th at **01:00 PM PDT**. Please reach out if you need more clarification or if you have further questions regarding the paper!
>
> Best regards.

---

### Official Review · Reviewer_cssb · 2023-07-21
**Review for "Objaverse-XL: A Colossal Universe of 3D Objects"**

**Rating:** 6
**Confidence:** 5

**Strengths:**

The contribution of a large object dataset provides a significant positive impact to a number of fields including, but not limited to, AI, graphics, computer vision, and robotics, as well as to fields where an understanding of how humans use objects is vital, e.g., occupational therapy and cognitive psychology. The broad comprehensive coverage of the data sources and the practices for curation is credible.

**Additional Feedback:**

None noted.

**Clarity:**

Though acronyms, e.g., CLIP, NSFW, and CAD are standard in the communities that use the corresponding assets, given the broader impact of a large object dataset, it is recommended to pre-expand the acronyms for broad accessibility of the paper. The same is recommended for the first introduction of datasets such as ABO and GSO (GSO can be abbreviated later in the Experiments section), and for algorithms such as SAM.

It is recommended to avoid the use of informal language in a technical paper, e.g., 'ginormous' in the teaser. If the intent is to convey the scale of the dataset, the word 'colossal' suffices, or a quantitative value (e.g., 10+ million) can be more informative.

Similarly, it is recommended avoid non-quantifiable descriptors, such as "very" in "very strong" (line 22), "very specialized" (line 29), "often" in "often still" (line 35). It is also recommended to avoid temporally anchoring language, e.g., "This year" (line 65).

 like --> such as (line 25, 42)

Line 47: "and is" --> ". Objaverse-XL is"

Figure 3: its color --> their color

**Correctness:**

In several locations, the paper uses the term "significant", however, the submission lacks any testing for statistical significance.

E.g., in line 50 it is claimed that Zero123 shows "significantly better" zero-shot generalization when pre-trained with Objaverse-XL (presumably when compared to training using Objaverse 1.0 as discussed in line 210). However, there has been no statistical testing to support the claim of significance. Figure 5 reiterates the claim of significant improvement without any quantitative support for significance. In the case of comparing to Zero123 trained with Objaverse 1.0, no quantitative results appear to be provided to enable conduction of statistical testing. If the reason is because of the use of in-the-wild images where quantitative ground truth is unavailable, then the claim that the results are significantly better has to be altered to a more accurate characterization of the results.

In the case of GSO, Table 2 once again makes the claim that Zeroshot123 fine-tuned on a high-quality subset of Objaverse-XL "significantly outperforms" the base model trained on the whole of Objaverse-XL, without any statistical testing for significance.

In the case of PixelNeRF, line 243 claims that generalization is "significantly better". However, the metric for measuring generalization has not been clarified. In Table 2, the claim of "significant performance improvement" is not supported via statistical testing. Also, it is not clear why the standard deviation has only been reported for the model trained with Objaverse-XL and not for the base.

If significance is to be reported, then the experiments have to be conducted properly to correctly report conclusions from the significance testing. If a statistical test appropriate to the data does not permit the null hypothesis to be rejected, then the differences between results without and with Objaverse-XL cannot be automatically assumed to be significant.


In Section 3(d) of the checklist in the supplementary, readers are directed to the Appendix to look for details on amount and type of compute used. However, the appendix (Section 6, lines 483-525 of the supplementary) lack details on the compute resources. Though training details on hyperparameters are provided in Section B, no details are provided on the type and number of compute resources used.

**Documentation:**

The dataset itself has not been shared for reviewers to view as per the submission instructions at https://nips.cc/Conferences/2023/CallForDatasetsBenchmarks:

>>>
If the dataset can only be released later, you must include instructions for reviewers on how to access the dataset. This can only be done after the first submission by sending an official note to the reviewers in OpenReview.
<<<

[If an official note was sent out, e.g., via the Notifications, this reviewer unfortunately did not receive.]

In the appendix, it is stated that the dataset will be shared via Hugging Face. However, given the scale of the data, and the lack of details on the dataset itself, it would have been beneficial to view the dataset to understand its content and nature of the differences from Objaverse 1.0.

**Ethics:**

The source and/or permissions of the in-the-wild images used in Figure 5 have not been discussed. If the source can be clarified, then the paper does not need to go to an ethics reviewer.

**Limitations:**

The negative societal impacts of the work have not been discussed. Though the checklist in the supplementary (Section 1(c), line 450) directs readers to the Appendix for discussion of potential negative societal impacts, upon perusal of the Appendix (lines 483--525 of the supplementary document), there was no information identified on negative societal impacts.

Suggestions on potential negative societal impacts: A curated large object dataset such as Objaverse-XL with realistic geometries and textures enables consumer-oriented applications such as 3D printing, or paired with inverse rendering and/or multi-view photorealistic rendering approaches, enables image manipulation with increasing accuracy, as demonstrated in the paper. The 3D printing of nefarious objects such as articles enabling damage with increasing accuracy can lead to intentionally harmful societal impact. Similarly, the manipulation of images to present information from a different viewpoint or to place 3D content within the hands of unintended user, can have a negative impact on the mental state of the viewer or present a harmful reinterpretation of the unmanipulated content. The paper should consider a discussion of specific examples relevant from the dataset for the above points.

**Opportunities For Improvement:**

The exposition lacks a multi-lens view into the dataset as compared to prior datasets such as Objaverse-1.0 or ShapeNet. Though histograms of polygon, vertex, and edge count are provided, their graphs are (quite naturally) similar to each other, and no analysis is provided of what different parts of the histograms allude to. A more comprehensive breakdown of object statistics in terms of geometry and appearance (where available) will be informative to comprehend the scale of the object diversity in relationship to prior datasets. For instance, though it is clear that there are several orders of magnitude more objects than prior datasets, and though the experiments provide indirect lenses into the data diversity, there are opportunities to provide direct lenses as well. As an example, summaries on metrics such as object volume, aspect ratio, curvature, and convexity serve as sources of direct information on the object geometry. Also, from the description, the distribution over categories such as room versus objects or CAD models versus 3D scans, and the relationship of those distributions to Objaverse 1.0 is not clear.

Considering the benefit of a large object dataset, the discussion of intended use cases (line 782-784) appears to be shallow. It will be instructive for future researchers for the submission to contain a more comprehensive discussion of the ways in which Objaverse-XL has and can be used.

Concerns are noted on the use of the term 'significance' in multiple portions of the paper. These concerns are further discussed in the "Correctness" section.

**Relation To Prior Work:**

The difference from prior contributions in terms of dataset size and impact on downstream tasks has been clearly elucidated, however, as discussed in the section on "opportunities for improvement", differences in terms of metrics pertaining to the objects themselves can be more comprehensively discussed.

**Summary And Contributions:**

The paper contributes a colossal object dataset, Objaverse-XL with 10 million 3D objects, a >10fold increase compared to the existing large object dataset Objaverse 1.0. The paper also contributes results of zero-shot novel view synthesis obtained by training using Objaverse-XL, and compared to prior models trained using Objaverse 1.0.

[Score raised from 4 to 6 after reviewer/author discussion comments]

---

> ### Author Response · Authors · 2023-08-23
> **Author Response**
>
> We thank the reviewer for their review. We are glad that they found the dataset provides a “significant positive impact to a number of fields including, but not limited to, AI, graphics, computer vision, and robotics, as well as to fields where an understanding of how humans use objects is vital, e.g., occupational therapy and cognitive psychology” and that the “practices for curation is credible.”
>
> > Avoid the use of informal language in a technical paper.
> >
>
> Thank you for your suggestion. We have updated the teaser figure to replace ginormous with 10+ million, and also updated the title of the paper from “Objaverse-XL: A Colossal Universe of 3D Objects” to “Objaverse-XL: A Universe of 10M+ 3D Objects”. We agree that the changes make these key parts of the paper more informative.
>
> > The paper uses the word “significant” but lacks testing for statistical significance.
> >
>
> Thank you for pointing this out. We have added statistical significance testing to Appendix F and G, which reports statistical significance for each of the quantitative evaluations, and also added a new human evaluation for the Zero123-XL results.
>
> Statistical significance tests were conducted on the Zero123-XL and PixelNeRF models using the paired *t*-test. On all tests, models trained with Objaverse-XL showed statistically superior performance compared to non-Objaverse-XL baselines, with results deemed significant at a p-value of less than 0.05. The comparisons involved PixelNeRF experiments with models pre-trained on 2 million Objaverse-XL objects and evaluated on ShapeNet and DTU, and Zero123-XL experiments on Google Scanned Objects in a 0-shot setting.
>
> For a more complete evaluation, we have also conducted a human evaluation that asks users to vote if they prefer the novel view synthesis generated from Zero123-XL compared to the baseline Zero123 model (trained on Objaverse 1.0). Users preferred the view synthesis of Zero123-XL over the baseline Zero123 model 62.19% of the time. The results have a z-score of 4.24, surpassing the critical value of 1.96, indicating that there is a statistically significant difference in preferences between the models' outputs. We have added a section on the Zero123-XL human evaluation to the appendix.
>
> > For pixelNeRF, why is the standard deviation only reported for the model trained with Objaverse-XL and not for the baseline?
> >
>
> We have updated the table to better clarify that the baseline model comes directly from the PixelNeRF paper and that we’re reiterating what they report. Nevertheless, we believe our statistical significance tests that show the paired *t*-test results provide a much better indication of statistical significance for these results.
>
> > Can you provide more dataset analysis?
> >
>
> Thanks for the suggestion! We have incorporated more analysis and insights of the dataset into the paper in Section J of the appendix. In particular, we show several more distributions of properties of objects captured in the metadata during rendering, such as plots with respect to the size of the object’s hierarchy, the number of materials, and the aspect ratios.
>
> The focus on this dataset was in scaling the number of objects, and thus, there are very few additional scenes beyond those present in Objaverse 1.0. For GitHub, the scenes are often stored in file types such as `.unity` or `.umap` (which we do not include because they can only be opened with proprietary software). But, the scene files utilize many of the objects that we’ve collected (e.g., by linking to individual `.fbx` objects, which we include). Thingiverse and the Smithsonian Institute have very few scenes, and Polycam has on the order of 10K scanned scenes. Moreover, the objects coming from Polycam and the Smithsonian Institute are all scanned objects, while those that come from GitHub and Thingiverse are nearly all 3D modeled objects.

---

> > ### Author Response · Authors · 2023-08-23
> > **Author Response (Part 2)**
> >
> > > Discussion of societal impacts.
> > >
> >
> > We have added a section on the societal impacts of Objaverse-XL and Zero123-XL to the appendix.
> >
> > We believe that models such as Zero123-XL, and those trained on Objaverse-XL, will enhance the ease of 3D content creation, enabling broader accessibility for individuals and businesses to participate. With the rise of mixed-reality glasses such as Apple's Vision Pro, the demand for 3D content is surging. Creating 3D content with tools such as Blender and Maya is time consuming and difficult, with tasks such as modeling a mere sofa requiring a day's effort from an expert 3D artist. We hope that by making 3D development faster and more accessible, it will enable many future applications in robotics simulation, AR/VR, games, education, architecture, manufacturing, healthcare, medicine, and biology, among many other industries.
> >
> > Openly accessible datasets, such as Objaverse-XL, serve as a crucial component for the democratization of knowledge to the AI community. Such resources allow others to access and utilize tools that might have been previously reserved for a select few organizations. Additionally, openly available datasets make it much easier for the AI community to verify the merits of different models and training algorithms, as it enables different organizations to use the same training data. Moreover, The inherent transparency of openly available datasets cultivate a foundation of trust. As these datasets are open to scrutiny, they invite collective oversight, ensuring that the methods and sources of data collection are held to high standards.
> >
> > However, we also acknowledge that there are several potential negative societal impacts with the release of Objaverse-XL. One concern is the development hyper-realistic fake content, such as deepfakes, which can be harnessed for misinformation campaigns or identity theft. The automation capabilities of AI may diminish the demand for human roles in areas like 3D modeling, subtly altering the employment landscape. An over-reliance on AI for generating 3D objects may also lead to an oversaturation of similar designs, potentially suppressing human creativity. Lastly, despite the democratizing intent of openly accessible datasets, the primary beneficiaries might often be those with more substantial resources or technological infrastructure that can launch large-scale training runs, inadvertently perpetuating economic disparities.
> >
> > > Can you expand on discussion of possible use cases?
> > >
> >
> > Yes, we have expanded on the intended use cases of the dataset in the datasheet, and highlighted.
> >
> > In particular, we mention that a key use case of Objaverse-XL, and models such as Zero123-XL, is for making 3D content creation more accessible through generative modeling. We hope that by enabling a much wider audience to build content in 3D, it will enable many future applications. Beyond generation, Objaverse 1.0 has enabled many exciting research directions. We believe that the strong scaling trends exhibited in our paper suggest that Objaverse-XL will better empower similar tasks. In particular, we are excited to see Objaverse-XL be used for building on works that use Objaverse 1.0, such as in building 3D-LLMs [[1](https://arxiv.org/abs/2307.12981)], 3D retrieval systems [[2](https://arxiv.org/abs/2305.08275), [3](https://arxiv.org/abs/2305.10764)], representation learning on synthetic images [[4](https://arxiv.org/abs/2308.03977)], text-to-texture generation [[5](https://arxiv.org/abs/2306.06212), [6](https://arxiv.org/abs/2303.11396)], and robotic simulation and robustness [[7](https://arxiv.org/abs/2212.08051)], among others.
> >
> > > Expand acronyms, such as CLIP, NSFW, and CAD.
> > >
> >
> > Thank you for the suggestion. We have revised the paper such that the first time relevant abbreviations are used, we explicitly write its full phrase to make it more clear.
> >
> > > Gramatical corrections.
> > >
> >
> > Thank you for your suggestions. We have revised the draft with your updates.
> >
> > > What type and and how much compute resources were used?
> > >
> >
> > Training the Zero123-XL model was done on 256 NVIDIA A100s over the course of 4 days, for a total of around 25K GPU hours. Training the PixelNeRF model was done with 8 NVIDIA A100's for 2 days, totaling 384 GPU hours. Rendering the objects was done for around 1 week on 48 NVIDIA T4 GPUs, totaling around 8K GPU hours. Both training and rendering were conducted using AWS. We have added these details to the supplementary material.

---

> > > ### Author Response · Authors · 2023-08-23
> > > **Author Response (Part 3)**
> > >
> > > > The source of the in-the-wild images.
> > > >
> > >
> > > The source of the in-the-wild images comes from images generated from a text-to-image model (Midjourney v5 in this case) or from various sources on the web, which is also what was done in [Zero123](https://arxiv.org/abs/2303.11328) and other notable computer vision papers that test for in-the-wild generation (e.g., [GPT-4](https://arxiv.org/abs/2303.08774), [Flamingo](https://arxiv.org/abs/2204.14198)). We believe the use of these images is reasonably supported by fair use, as the use of the images is non-commercial, for informative educational research purposes, makes up a small minority of the paper, and does not diminish the original work’s value. However, we are happy to provide a link to the sources of the images upon publication.
> > >
> > > > What is the link for the dataset?
> > > >
> > >
> > > The dataset is available on Hugging Face at [https://huggingface.co/datasets/allenai/objaverse-xl](https://huggingface.co/datasets/allenai/objaverse-xl).
> > >
> > > We have also provided a download API and tutorial to get started working with it!
> > >
> > > We sincerely appreciate the detailed review and are looking forward to our discussion with you in the rebuttal phase. Please let us know if you have any additional questions!

---

> > ### Comment · Reviewer_cssb · 2023-08-29
> > **Details on statistical evaluation**
> >
> > Details on the statistical testing are still somewhat unclear.
> > 1. When conducting the statistical testing, was a test for normality conducted to warrant the use of the t-test?
> > 2. If pixelNeRF was re-evaluated on ShapeNet and DTU for the statistical testing, it is not clear why the mean from [75] is reported instead of the mean of the metrics obtained when re-evaluating pixelNeRF for this submission. Though one can understand that the submission seeks to report the prior paper's result as is, currently, the base case of results (i.e. without Objaverse-XL pretraining) used in statistical testing cannot be related to the first ow of Table 3.
> > 3. What are the sizes of the datasets used for evaluating ShapeNet and DTU (i.e., the amount of data on which statistical testing was conducted)? Same question for GSO, i.e., was the entire dataset used or part of it?

---

> > > ### Author Response · Authors · 2023-08-29
> > > **Response on statistical evaluation**
> > >
> > > We thank the reviewer for their response. Below, we address each of their questions regarding statistical significance.
> > >
> > > > When conducting the statistical testing, was a test for normality conducted to warrant the use of the t-test?
> > > >
> > >
> > > T-tests are well known to be robust to violations of normality (although their power may be lower in such settings). Quoting from [[1](http://rctdesign.org/techreports/arphnonnormality.pdf)], “it is widely but incorrectly believed that the t-test and linear regression are valid only for Normally distributed outcomes.” Nevertheless, we ran the D'Agostino's K-squared test and obtained p-values for the two samples of:
> > >
> > > - 0.6948, 0.7282 ShapeNet PSNR
> > > - .8533, .3171 ShapeNet SSIM
> > > - .6019, 1.015 ShapeNet LPIPS
> > > - .5797, 1.090 DTU PSNR
> > > - 0.5297, 1.27 DTU SSIM
> > > - 0.4291, 1.691 DTU LPIPS
> > >
> > > Thus, we fail to detect a significant departure from normality.
> > >
> > > [[1](http://rctdesign.org/techreports/arphnonnormality.pdf)] Lumley, Thomas, et al. "The importance of the normality assumption in large public health data sets." *Annual review of public health* 23.1 (2002): 151-169.
> > >
> > > > If pixelNeRF was re-evaluated on ShapeNet and DTU for the statistical testing, it is not clear why the mean from [75] is reported instead of the mean of the metrics obtained when re-evaluating pixelNeRF for this submission.
> > > >
> > >
> > > Thanks for the suggestion. When re-evaluating the pixelNeRF results, we obtained nearly identical results to what they report in their paper. Specifically, we obtain a DTU PSNR of 15.32 ± 1.38 and a ShapeNet PSNR of 22.77 ± 0.38, compared to the current reported means of 15.32 and 22.71, respectively. We will add these re-evaluated results with the standard deviations to the paper.
> > >
> > > > What are the sizes of the datasets used for evaluating ShapeNet and DTU (i.e., the amount of data on which statistical testing was conducted)? Same question for GSO, i.e., was the entire dataset used or part of it?
> > > >
> > >
> > > We use standard subsets of 90 objects for each of ShapeNet, DTU, and GSO. These datasets are commonly used for evaluation of novel view synthesis results in papers such as pixelNeRF, GenVS, and Zero123.
> > >
> > > We hope that the rebuttal clarifies the questions raised by the reviewer. We would be very happy to discuss any further questions about the work, and would really appreciate an appropriate increase in the score if the reviewer’s concerns are adequately addressed to facilitate acceptance of the paper.

---

> ### Author Response · Authors · 2023-08-28
> **Rebuttal Reminder**
>
> Hi Reviewer cssb,
>
> Just a quick reminder that the rebuttal deadline is **tomorrow**, August 29th at **01:00 PM PDT**. Please reach out if you need more clarification or if you have further questions regarding the paper!
>
> Best regards.

---

### Official Review · Reviewer_fkNJ · 2023-07-22
**The largest 3D dataset so far but may require more experiments on more tasks and baselines**

**Rating:** 7
**Confidence:** 4
**Clarity:** Yes. The paper is well written and ea…

**Strengths:**

- The paper presents the largest 3D dataset Objverse-XL, with 10.2M 3D models.
- Methods are designed to analyze and clean the 3D data.
- The paper demonstrates that more 3D data leads to better performance in view synthesis tasks.

**Additional Feedback:**

All feedback are are provided above

**Correctness:**

The dataset is constructed soundly and using some factors to clean the data such as NSFW annotations. The data license is also adequately explained.

**Documentation:**

The paper does not provide a URL to access the dataset, and the authors did not clarify whether the dataset and related toolchains would be open-sourced.

**Limitations:**

- The paper acknowledges that Objverse-XL is still significantly smaller than billion-scale image-text datasets. The authors may future provide more suggestions on how to scale up 3D datasets, as high-quality 3D data on the internet is much scarcer than image-text data..
- The paper may show some examples of low-quality models in Objverse-XL, as well as some failure cases after training.

**Opportunities For Improvement:**

Regarding the dataset:

- How many categories? And how are the data distributed among different categories?
- Is there more metadata available, such as text descriptions of the objects?
- More diversity and quality analysis on some important properties, such as topology, material, and texture.

Regarding the experiments:

- More baseline models and tasks should be used to provide more solid conclusions. For example,
    - Reconstuction or Neural Surface Reconstruction, which can demonstrate the geometric advantage of the dataset, such as NeuS, VolSDF, and Voxsurf.
    - 3D Generation, which is a more intuitive task for this dataset, such as GET3D and Shape-E.
- For the experiments, which kind of 3D data are more effective, and what kind of data should be added in the future? The paper gives some clues on `Alignment Finetuning` in sec.4.1, but the standard for high-quality is not clear. It will be insightful for the community if such questions are ablated and discussed in the paper.

**Relation To Prior Work:**

The paper does not compare Objverse-XL to some datasets, such as ScanObjectNN and AKB-48. Additionally, in L80-81, it is not appropriate to group GSO and OmniObject3D into CAD models, as they are scanned objects.

**Summary And Contributions:**

This paper presents the largest 3D dataset Objverse-XL, which contains 10.2M 3D models. The effectiveness of the dataset is demonstrated in training view synthesis models, i.e., Zero123 and PixelNeRF. The experiments show that using models trained with more 3D data as pretrained model achieves better generalization performance.

---

> ### Author Response · Authors · 2023-08-23
> **Author Response**
>
> We thank the reviewer for their review. We are glad they found the dataset is “constructed soundly”, that the paper is “well written and easy to follow”.
>
> > Is there more metadata available, such as text descriptions of the objects?
> >
>
> For each of the objects, we provide a large amount of metadata extracted from both the source website, from Blender, and from the renders.
>
> **********Metadata from the source websites.********** On each of the websites, we extract metadata for each of the objects. On Thingiverse, Polycam, Sketchfab, and Smithsonian Institute, each object has a textual name associated with it. On GitHub, the file name can be used as noisy text supervision. On Thingiverse, Polycam, Sketchfab, and GitHub, we also have access to some notion of how popular the object is, such as having access to the number of views or likes it has.
>
> ******CLIP Embeddings.****** We provide CLIP embeddings for each of the objects. The CLIP embeddings come from taking the mean normalized CLIP embeddings from 12 different renders around the object. The CLIP embeddings can be used for language-search retrieval, image-search retrieval, and for filtering different subsets of the dataset.
>
> **Metadata from Blender**. While rendering the objects, we extract a large amount of metadata from Blender, including:
>
> - `fileIdentifier`: A unique web identifier of the 3D object, such as its URL.
> - `sha256`: The sha256 hash of the contents of the file. The file hash can be used for (1) deduplication and (2) to prevent data poising attacks by ensuring users are downloading the same objects that we intended (see [here](https://arxiv.org/abs/2302.10149) for more).
> - `license`: The license of the individual 3D object.
> - `animationCount`: The number of animations available on the object.
> - `armatureCount`: The number of armature annotations in the scene. An armature annotation outlines the skeleton bone structure of an object (such as a person or an animal) that can be used to easily animate it. For example, if a human object has an armature annotation, it can be deformed with any of the Adobe [Mixamo](https://www.mixamo.com/#/) animations.
> - `fileFormat`: The format of the 3D file.
> - `materialCount`: The number of materials on the 3D object.
> - `fileSize`: The size of the file (in bytes). Note that the size of linked files is not included, such as texture files if they’re linked instead of embedded.
> - `polygonCount`: The number of polygons on the 3D object.
> - `vertexCount`: The number of vertices on the 3D object.
> - `edgeCount`: The number of edges on the 3D object.
> - `objectCount`: The number of objects in the hierarchy of the scene.
> - `lampCount`: The number of lamp lights that are included in the 3D object file.
> - `meshCount`: The number of objects in the hierarchy that have a mesh component to them.
> - `shapeKeyCount`: The number of shape keys on the object, which are used to deform objects into new shapes for animation.
> - `linkedFileCount`: The number of linked files identified on the object, such as links to different texture files that might be part of a larger GitHub repository.
> - `boundingBoxVolume`: The volume of the bounding box of the object.
> - `boundingBox`: The bounding box of the 3D object (which includes its minimum and maximum x, y, and z values).
>
> > How many categories are in the dataset?
> >
>
> Obtaining the number of categories for internet-scale datasets, such as LAION-5B or web-text crawled from Common Crawl, is often infeasible. Here, most items cannot be cleanly mapped to a set of categories that is universally useful for retrieval or filtering. The same thing is true with Objaverse-XL, where the vast majority of objects cannot be cleanly mapped to a set of predefined categories that are universally useful.
>
> Instead, for use cases such as retrieval or filtering, similarity search is often what is more practical. Here, one can leverage embeddings of the objects (e.g., CLIP embeddings of rendered images of the objects), and metadata for filtering, to retrieve objects based on a specific use case. In this work, we already demonstrate two specific use cases that such an approach works well on: (1) finding our alignment fine-tuning subset of objects and (2) finding objects that are aesthetically pleasing. The metadata can be used to filter objects by popularity on the source website (e.g., likes on Sketchfab) or to have a minimum polygon count cutoff threshold, for example. The CLIP embeddings can be used to retrieve objects from either an image similarity search or from a text description.
>
> > Can you provide more dataset analysis?
> >
>
> Yes, we have incorporated more analysis of the dataset into the paper in Section J of the appendix. In particular, we show several more distributions of properties of objects captured in the metadata during rendering, such as plots with respect to the size of the object’s hierarchy, the number of materials, and the aspect ratios.

---

> > ### Author Response · Authors · 2023-08-23
> > **Author Response (part 2)**
> >
> > > What are suggestions on how to scale up 3D datasets?
> > >
> >
> > Scaling up 3D datasets beyond the scale of Objaverse-XL could involve scaling up the dataset collection effort as demonstrated in Objaverse (e.g., by re-crawling sources like GitHub or adding new sources), from additional 3D dataset forms such as scans/captures (e.g., via NeRF), or generative methods (e.g., via DreamFusion). Additionally, as we mention in the paper, better conversion tools to convert the many 3D file formats into formats that can be read by Blender has the potential to add several million new 3D objects to the dataset.
> >
> > > More generation baseline results and tasks should be used to provide more solid conclusions (such as training the following models on Objaverse-XL: NeuS, VolSDF, Voxsurf, GET3D, and Shap-E).
> > >
> >
> > Thank you for the suggestion. We have added Section H to the appendix, which shows new generation results. In particular, we show how we can use Zero123-XL to obtain strong results for both single image → 3D generation and text → 3D generation. We’ve added our implementation into [Threestudio](https://github.com/threestudio-project/threestudio#zero-1-to-3-), a popular framework for 3D content generation, to allow others to generate models as well and use it as a baseline.
> >
> > In this paper, we focus on constructing Objaverse-XL as a large-scale open dataset of 3D objects, training Zero123-XL to serve as a foundation model for 3D research, and show strong scaling trends for both Zero123-XL and PixelNeRF when scaling up to use Objaverse-XL. We believe the scaling trends indicate that many other modeling approaches can leverage Objaverse-XL for training. But, to ablate the pre-training experiments on many different modeling approaches, it would require a considerable amount of GPU time (~25K GPU hours per experiment), and so we needed to focus on experiments we were most confident in.
> >
> > > What kind of data is useful for alignment fine-tuning?
> > >
> >
> > For alignment fine-tuning, the data that is used depends on the use case. Objaverse-XL has many diverse objects, so for pre-training, we train Zero123-XL on everything to learn a strong shape prior. Then, because we are most interested in novel view synthesis and generative 3D modeling of real world objects, we make the alignment fine-tuning data most similar to the distribution of real world objects. In particular, we filter by different tags from the source webpages, by poly count, by the number of textures and materials on the object, and by the 3D aspect ratio (to not include objects that are mostly flat). We find that this distribution tends to be more similar to the distribution of real world objects than Objaverse-XL as a whole, and after fine-tuned on this subset, it makes the generations more realistic. However, other applications may want to generate low-poly objects or objects under a cartoonized style, for example, so the alignment fine-tuning subset would be different for the different use cases.
> >
> >
> > > The paper may show some examples of low-quality models in Objaverse-XL, as well as some failure cases after training.
> > >
> >
> > Thank you for the suggestion. We have added Section I to the appendix, which discusses possible sources of undesirable objects, including those with scanning artifacts, low-poly objects, textureless objects, and objects that appear on a surface. We also discuss several failure cases for Zero123-XL, including counting artifacts, difficulties with smudging faces, occasional unclosed objects when viewed from the bottom, and relative positional issues when working with multiple objects.
> >
> > > Pointers to cite AKB-48 and ScanObjectNN in the related work.
> > >
> >
> > Thanks for providing pointers to related work! We have added ScanObjectNN, which includes 15,000 objects across 15 categories, and AKB-48, which includes 2,037 articulated scanned object models across 48 categories, to the related work. We have also better separated prior 3D datasets into those that are scanned compared and those that are CAD models.
> >
> > We have additionally better clarified that GSO and OmniObject3D include scanned objects.
> >
> > > What is the link for the dataset?
> > >
> >
> > The dataset is available on Hugging Face at [https://huggingface.co/datasets/allenai/objaverse-xl](https://huggingface.co/datasets/allenai/objaverse-xl).
> >
> > We have also provided a download API and tutorial to get started working with it!

---

> > > ### Author Response · Authors · 2023-08-23
> > > **Author Response (Part 3)**
> > >
> > > > Will the dataset and code be open-sourced?
> > > >
> > >
> > > Yes, the dataset is openly available, we provide an open-source API to access the dataset, and will provide an open source package for rendering and preprocessing the objects to make it easy for the community to use. Additionally, the weights for Zero123-XL are openly available [here](https://zero123.cs.columbia.edu/assets/zero123-xl.ckpt) and can be directly loaded into the [Zero123](https://github.com/cvlab-columbia/zero123) training repository.
> > >
> > > Please let us know if you have any additional questions.
> > >
> > > Finally, if you believe this paper should be highlighted at the conference, could you please consider raising your score to reflect that?

---

> ### Author Response · Authors · 2023-08-28
> **Rebuttal Reminder**
>
> Hi Reviewer fkNJ,
>
> Just a quick reminder that the rebuttal deadline is **tomorrow**, August 29th at **01:00 PM PDT**. Please reach out if you need more clarification or if you have further questions regarding the paper!
>
> Best regards.

---

> > ### Comment · Reviewer_fkNJ · 2023-08-29
> >
> > 1. Thanks for providiing detailed meta information. It would be helpful if it could be included in the paper along with examples demonstrating how to utilize this information.
> >
> > > The same thing is true with Objaverse-XL, where the vast majority of objects cannot be cleanly mapped to a set of predefined categories that are universally useful.
> >
> > 2. I agree that CLIP-based retrieval or filtering can contribute to selecting a high-quality subset, but I believe that the scale of objverse is not yet comparable to LAION-5B. It might be more suitable to compare it to OpenImage, which has 20 million images with human-verified label as training data \[1\].
> >
> > 3. Regarding the failed examples, the community might be interested in knowing the quantity of such examples and the actual amount of valid data within the 10+ million samples. In addition to the whole valid dataset, as the authors mentioned, subset for alignment is also critial and it is task-dependent. Therefore, it may be necessary to examine the available data for specific tasks. The paper utilized 1.3 million samples for Zero123-XL, but it is unclear how many examples would be available for other common downstream tasks such as Reconstruction or Neural Surface Reconstruction (NeuS, NeUDF etc.), or 3D generation (GET3D, Shape-E, etc.).
> >
> > > we train Zero123-XL on everything to learn a strong shape prior.
> >
> > 4. The authors mentioned learning a strong shape prior, and it would be beneficial to present some experiments showcasing this. Even in the 3D Reconstruction section of the appendix, geometric information without textures is not demonstrated.
> >
> > [1] https://github.com/openimages/dataset/blob/main/READMEV3.md#image-level-labels

---

> > > ### Author Response · Authors · 2023-08-29
> > >
> > > We thank the reviewer for their response. Below, we address each of their remaining comments.
> > >
> > > > Thanks for providing detailed meta information. It would be helpful if it could be included in the paper along with examples demonstrating how to utilize this information.
> > > >
> > >
> > > Thanks for the suggestion! We will add documentation on how to access the metadata information and point to it in the paper.
> > >
> > > > It would be useful to add categories to the objects
> > > >
> > >
> > > Thanks for the suggestion. While we still do not believe all objects can cleanly be categorized, there might be a subset of objects that are mappable to categories, and thus be useful for some applications. However, categories are not used in the majority of tasks used by the community for Objaverse 1.0, which most often use renders of the objects, point clouds, and text descriptions. Moreover, the category annotations were not needed for our Zero123-XL, pixelNeRF, and 3D generation experiments. So, we leave the use of categories to be explored in future work.
> > >
> > > > Regarding the failed examples, the community might be interested in knowing the quantity of such examples and the actual amount of valid data within the 10+ million samples.
> > > >
> > >
> > > Thanks for the suggestion. Within the 10+ million samples, we only keep objects that are “valid” 3D objects (i.e., objects that can be loaded into Blender and are not empty). Computing the quantities of possible undesirable examples in the dataset requires reprocessing through the entire dataset, which will take a considerable amount of computing resources. We will try our best to do this for the camera-ready submission!
> > >
> > > > In addition to the whole valid dataset, as the authors mentioned, subset for alignment is also critial and it is task-dependent. Therefore, it may be necessary to examine the available data for specific tasks. The paper utilized 1.3 million samples for Zero123-XL, but it is unclear how many examples would be available for other common downstream tasks such as Reconstruction or Neural Surface Reconstruction (NeuS, NeUDF etc.), or 3D generation (GET3D, Shape-E, etc.).
> > > >
> > >
> > > For our alignment fine-tuning subset used for Zero123-XL training, we focused on obtaining a subset of data that is realistic towards objects in the real world. We believe this same subset would already be useful for many similar tasks, such as those that you mention: neural surface reconstruction and 3D generation. For tasks that have a different goal, they might require a different alignment fine-tuning subset of data. For example, if 3D generation of cartoonish objects (instead of real-world objects) was desirable, that would likely benefit from a different alignment fine-tuning subset.
> > >
> > > Moreover, we reiterate that Zero123-XL can be used to obtain strong 3D generation results, as we demonstrate in Appendix H and implement in [threestudio](https://github.com/threestudio-project/threestudio#zero-1-to-3-), a popular GitHub repository for 3D generation.
> > >
> > > > The authors mentioned learning a strong shape prior, and it would be beneficial to present some experiments showcasing this. Even in the 3D Reconstruction section of the appendix, geometric information without textures is not demonstrated.
> > > >
> > >
> > > A shape prior can be demonstrated using objects with textures, which is often much harder than demonstrating it on objects without textures. Here, the synthesized image not only has to predict the shape, but also the lighting and texture of the object from a different viewpoint. We demonstrate many pages of examples of novel views synthesized from Zero123-XL in the appendix. Nevertheless, we would be happy to add more examples of novel views synthesized on objects without textures for the camera-ready submission.
> > >
> > > We hope that the rebuttal clarifies the questions raised by the reviewer. We would be very happy to discuss any further questions about the work, and would really appreciate an appropriate increase in the score if the reviewer’s concerns are adequately addressed to facilitate acceptance of the paper.

---

### Official Review · Reviewer_7tXF · 2023-07-26
**A huge 3D object dataset**

**Rating:** 7
**Confidence:** 4
**Correctness:** Yes, the dataset is constructed in a …
**Clarity:** Yes.

**Strengths:**

This article introduces Objaverse-XL, which is comprised of 10.2M 3D assets and is at least an order of magnitude larger than existing 3D datasets, such as Objaverse 1.0 and ShapeNet, with the latter being two orders of magnitude smaller. The scarcity of 3D data has been a bottleneck hindering the development of 3D computer vision. The emergence of this dataset fills the gap of large-scale 3D datasets and provides strong support for the development of 3D computer vision. Furthermore, through experiments in the task of zero-shot novel view synthesis, it has been demonstrated that Objaverse-XL can enhance the performance of existing generation models, such as Zero123 and PixelNeRF.

**Additional Feedback:**

This is an exciting work that opens up new possibilities for future 3D research. With the current popularity of multimodal research, such as text-to-image generation work that has brought significant value to creators, there is great potential for further advancement. However, the article notes that the dataset only uses file names as textual descriptions for 3D assets, leaving open the question of whether there are methods to obtain more refined textual descriptions. Finding ways to obtain more detailed textual descriptions could potentially create even more research opportunities for the community.


**Documentation:**

No. The author team has not provided the dataset link.

**Limitations:**

Yes, the authors adequately addressed the limitations and potential negative societal impact.

**Opportunities For Improvement:**

While the proposed 3D object dataset is significantly larger than existing 3D datasets, there is still a gap in scale compared to 2D image datasets. The paper only demonstrates through experiments that this dataset can enhance the performance of existing generation models, such as Zero123 and PixelNeRF. However, these results alone are not sufficient to fully demonstrate the potential impact of this dataset in other areas. Therefore, further research is needed to explore whether this dataset has potential value for other computer vision applications.

**Relation To Prior Work:**

Yes.

**Summary And Contributions:**

This article presents a massive 3D object dataset, Objaverse-XL, crawled from several online resources, which contains a vast number of 3D object models of various types. Through experiments, it has been demonstrated that pre-training on this dataset can significantly improve the performance of existing 3D generation models, thereby providing potential opportunities for large-scale training in new 3D applications.

---

> ### Author Response · Authors · 2023-08-23
> **Author Response**
>
> We thank the reviewer for their review. We are glad they found our paper is “exciting work that opens up new possibilities for future 3D research” and shared the sentiment that “the scarcity of 3D data has been a bottleneck hindering the development of 3D computer vision.”
>
> > There is still a gap in scale compared to 2D image datasets.
>
> While there's a gap in scale when comparing our 3D dataset to existing 2D image datasets, it's essential to recognize the unique attributes and value each brings to the table.
>
> **2D and 3D datasets are complementary.** We believe that there's an opportunity for 2D and 3D datasets to complement each other rather than compete. With Zero123-XL, we already demonstrate how we can leverage both the sheer scale of pre-training a model on 2D image datasets (i.e., Stable Diffusion trained on LAION-5B) to learn many visual concepts, and then we can fine-tune that model on Objaverse-XL to endow a 3D understanding into the model. By leveraging both images and 3D objects, we get the best of both worlds and achieve strong generalization results to a long-tail distribution of objects that Zero123-XL works well on out “in-the-wild”.
>
> **3D provides more supervision than 2D.** A single 3D model is often more useful than a single 2D image, as 3D models encapsulate multi-angled perspectives of an object, allowing for a more comprehensive understanding of its structure and geometry. In this paper, each 3D model serves as the source of 12 distinct training images, rendered from various viewpoints. Work such as [Shap-E](https://arxiv.org/abs/2305.02463) have even demonstrated the efficacy of using up to 60 rendered views for each 3D object. Furthermore, with 3D models, it’s easy to generate point clouds, change the lighting, obtain depth maps, obtain segmentation, place objects together with other objects in a scene, and use the many animations and skeletal poses annotated on the objects to perform different types of tasks.
>
> > Further research is needed to explore whether this dataset has potential value beyond generation.
> >
>
> While we focus on the dataset in this paper, and provide considerable generation experiments, study scaling trends, and build Zero123-XL to serve as a foundation model for 3D research, we agree that this dataset has a significant amount of value beyond generation. Since the release of Objaverse 1.0, it has enabled a tremendous amount of exciting research, and we expect each of these applications to similarly be able to benefit from Objaverse-XL. In particular, we are excited to see Objaverse-XL be used for building on works that use Objaverse 1.0, such as in building 3D-LLMs [[1](https://arxiv.org/abs/2307.12981)], 3D retrieval systems [[2](https://arxiv.org/abs/2305.08275), [3](https://arxiv.org/abs/2305.10764)], representation learning on synthetic images [[4](https://arxiv.org/abs/2308.03977)], text-to-texture generation [[5](https://arxiv.org/abs/2306.06212), [6](https://arxiv.org/abs/2303.11396)], and robotic simulation and robustness [[7](https://arxiv.org/abs/2212.08051)], among other applications in AR/VR, multimodal learning, and generation. We believe that the strong scaling trends exhibited in our paper suggest that Objaverse-XL will similarly be useful for many other tasks.
>
> > What is the link for the dataset?
> >
>
> The dataset is available on Hugging Face at [https://huggingface.co/datasets/allenai/objaverse-xl](https://huggingface.co/datasets/allenai/objaverse-xl).
>
> We have also provided a download API and tutorial to get started working with it!

---

> > ### Author Response · Authors · 2023-08-23
> > **Author Response (Part 2)**
> >
> > > Finding more detailed textual descriptions could open up new research opportunities.
> >
> > We agree that obtaining more detailed textual descriptions is useful, and believe this is an exciting direction of current research. Indeed, the utilization of object names from sites like Sketchfab, Thingiverse, Polycam, and the Smithsonian Institute, and the use of file names from GitHub, is a preliminary step that serves as noisy labeled supervision for the textual description of the object.
> >
> > Recognizing the potential benefits, obtaining more detailed textual descriptions is an exciting current area of research, with recent works like [Cap3D](https://arxiv.org/abs/2306.07279) providing a nice approach. [Recent work](https://arxiv.org/abs/2307.10350) has also shown that using captions from image captioning models, as opposed to alt-text, enhances vision-language model performance until about 128M captions, after which generated caption diversity becomes a concern. Consequently, we believe that generated captions could significantly benefit Objaverse-XL, marking a promising research direction.
> >
> > Additionally, for many purposes, it may be possible to condition on text input, without using text input during training. For example, for generation, image-to-3D might be sufficient to perform text-to-3D, by first performing text-to-image with an off the shelf model, and then performing image-to-3D. We have added more details about the strong image-to-3D results obtained using Zero123-XL in Appendix H, and a demonstration of going from text-to-3D using text-to-image-to-3D. Our code and implementation is available in [threestudio](https://github.com/threestudio-project/threestudio#zero-1-to-3-).
> >
> > Please let us know if you have any additional questions!
> >
> > Finally, if you believe this paper should be highlighted at the conference, could you please consider raising your score to reflect that?

---

> ### Author Response · Authors · 2023-08-28
> **Rebuttal Reminder**
>
> Hi Reviewer 7tXF,
>
> Just a quick reminder that the rebuttal deadline is **tomorrow**, August 29th at **01:00 PM PDT**. Please reach out if you need more clarification or if you have further questions regarding the paper!
>
> Best regards.

---

### Author Response · Authors · 2023-08-23
**Overall Response**

We thank the reviewers for their thoughtful feedback on the work. We are encouraged to hear their positive feedback and that they find the dataset **provides a significant positive impact to a number of fields** including, but not limited to, AI, graphics, computer vision, and robotics, as well as to fields where an understanding of how humans use objects is vital, e.g., occupational therapy and cognitive psychology (Reviewer cssb), the work **fills the gap** of having a large-scale 3D dataset (Reviewer 7tXF26), that **the scarcity of 3D data has been a bottleneck** hindering the development of 3D computer vision (Reviewer 7tXF), that they find the work **exciting** and that it opens up **new possibilities for future 3D research** (Reviewer 7tXF26, Reviewer toL1), is **well-written** and easy to follow (Reviewer fkNJ), and that we **ensure the usability** of the dataset (Reviewer QWh5).

We have additionally made several revisions to the draft. In particular, we changed the title to be more descriptive, conducted a human evaluation of the Zero123-XL results, added a section on using Zero123-XL to perform Image → 3D and text → 3D generation, performed statistical significance testing on all quantitate results to show that the results training with Objaverse-XL are statistically significant compared to the baselines, added more analysis of the dataset, updated figures for clarity, and addressed several other comments mentioned by the reviewers. The changes to the revision are highlighted in red.

We now address each of the reviewers' comments below. Please let us know if you have any other questions and we would be happy to discuss!

---

### Decision · Program_Chairs · 2023-09-22

**Decision:**

Accept (Poster)

**Comment:**

This paper presents a new dataset of 10M 3D object assets which is the largest and much larger than ShapeNet/Objaverse 1.0. All reviewers and AC acknowledge the significance of this dataset contribution to the fields of 3D AI, ML, vision, and graphics. The authors have done some benchmarking for many applications like novel view synthesis in the original paper submission and text-to-3D and image-to-3D during rebuttal. While presenting more dataset statistics, and analysis, and showing its use for more applications in more thorough ways is strongly encouraged to add as some of the reviewers have pointed out, AC and all reviewers have reached to a clear consensus on recommending an acceptance of this paper.